

# The Sensitivity of the Fitch Wind Farm Parameterization to a Three-Dimensional Planetary Boundary Layer Scheme

Alex Rybchuk[1,2], Timothy W. Juliano[3], Julie K. Lundquist[2,4], David Rosencrans[2,4], Nicola Bodini[2], and Mike Optis[2]

[1]Department of Mechanical Engineering, University of Colorado Boulder, Boulder, Colorado, USA
[2]National Renewable Energy Laboratory, Golden, Colorado, USA
[3]Research Applications Laboratory, National Center for Atmospheric Research, Boulder, Colorado, USA
[4]Department of Atmospheric and Oceanic Sciences, University of Colorado Boulder, Boulder, Colorado, USA

**Correspondence:** A. Rybchuk (alex.rybchuk@colorado.edu)

**Abstract.** Wind plant wake impacts can be estimated with a number of simulation methodologies, each with its own fidelity and sensitivity to model inputs. In turbine-free mesoscale simulations, hub-height wind speeds can significantly vary with the choice of a planetary boundary layer (PBL) scheme. However, the sensitivity of wind plant wakes to a PBL scheme has not been explored because, until now, wake parameterizations were only compatible with one PBL scheme. We couple the Fitch wind farm parameterization with the new NCAR 3DPBL scheme and compare the resulting wakes to those simulated with a widely used PBL scheme. First, we simulate a wind plant in a pseudo-steady state under idealized stable, neutral, and unstable conditions with two PBL schemes: MYNN and the NCAR 3DPBL. For these idealized scenarios, MYNN consistently predicts internal wakes that are 0.25–1.5 m s$^{-1}$ stronger than internal 3DPBL wakes. However, because MYNN predicts stronger inflow winds than 3DPBL, MYNN predicts average capacity factors that are as large as 13 percentage points higher than with the 3DPBL, depending on the stability. To extend this sensitivity study, we conduct a month-long case study with both PBL schemes centered on the Vineyard Wind 1 lease area in the mid-Atlantic United States. Under stable and unstable conditions averaged across the month, MYNN again predicts stronger internal waking—by about 0.25 m s$^{-1}$. However, again due to stronger plant inflow wind speeds in MYNN, the 3DPBL generates 4.7%–7.8% less power than MYNN in August 2020, depending on the turbine build-out scenario. Differences between PBL schemes can be even larger for individual instances in time. These simulations suggest that PBL schemes represent a meaningful source of modeled wind resource uncertainty; therefore, we recommend incorporating PBL variability into future wind plant planning sensitivity studies as well as wind forecasting studies.

*Copyright statement.* This work was authored in part by the National Renewable Energy Laboratory, operated by Alliance for Sustainable Energy, LLC, for the U.S. Department of Energy (DOE) under Contract No. DE-AC36-08GO28308. Funding provided by the U.S. Department of Energy Office of Energy Efficiency and Renewable Energy Wind Energy Technologies Office and by the National Offshore Wind Research and Development Consortium under Agreement No. CRD-19-16351. The views expressed in the article do not necessarily represent the views of the DOE or the U.S. Government. The U.S. Government retains and the publisher, by accepting the article for publication,





acknowledges that the U.S. Government retains a nonexclusive, paid-up, irrevocable, worldwide license to publish or reproduce the published form of this work, or allow others to do so, for U.S. Government purposes.

# 1 Introduction

## 1.1 Motivation and Research Question

While wind turbines are known to modify the local atmosphere, there is debate on how to best capture their impacts in numerical weather prediction (NWP) models. Turbines generate wakes (regions of slower winds and increased turbulence) that potentially reduce downstream wind speeds and power output for neighboring turbines. Lee and Fields (2021) summarize the large degree of uncertainty regarding the impact of wake-associated losses on annual energy production: Some estimates predict losses as low as 5%, whereas others have predicted losses as high as 40%. The uncertainty of individual wake-loss estimates has also been estimated to be 10%–40%. These losses and uncertainties incur significant financial impact on the wind industry, potentially translating to millions of U.S. dollars of economic benefits (Lee and Fields, 2021). Thus, when planning the development of a new wind plant, it is important to accurately predict wake impacts and minimize the uncertainty of those impacts.

Part of the uncertainty in NWP-modeled wakes stems from uncertainty associated with hub-height wind speeds outside of the influences of turbines. Hub-height winds in NWP simulations can be significantly sensitive to a number of model inputs, particularly the choice of a planetary boundary layer (PBL) scheme (Storm and Basu, 2010; Carvalho et al., 2012; Yang et al., 2013; Carvalho et al., 2014; Draxl et al., 2014; Olsen et al., 2017; Yang et al., 2017; Fernández-González et al., 2018; Yang et al., 2019; Optis et al., 2020). PBL schemes govern turbulent fluxes and mixing within the atmospheric boundary layer. At present, 13 different PBL schemes are available within the Weather Research and Forecasting (WRF, Skamarock et al., 2021) model, and there is no single-best PBL scheme for wind resource assessment. As just one example, Draxl et al. (2014) evaluated seven PBL schemes using measurements from a meteorological mast at the Høvsøre wind energy test site. They found that the optimal PBL scheme varies with stability: at this site, MYJ (Janjić, 1994) performed best under stable conditions, ACM2 (Pleim, 2007) performed best under neutral conditions, and YSU (Hong et al., 2006) performed best under unstable conditions. The wind speed differences between PBL schemes can be significant. For example, Optis et al. (2020) and Rybchuk et al. (2021) found that annually averaged 100-m winds off the coast of California could differ by as much as 1.75 m s$^{-1}$ depending on the choice of PBL scheme. While the sensitivity of hub-height winds to PBL scheme has been explored in turbine-free NWP simulations, the resulting impacts on wake simulations have not been explored. To the best of the authors' knowledge, to date, all mesoscale WRF simulations with explicitly represented wind turbines have been conducted with the MYNN PBL scheme (Nakanishi and Niino, 2009; Olsen et al., 2017).

While wake model sensitivity studies have not examined the impact of PBL schemes on mesoscale wakes, they have shown that NWP-modeled wakes can be sensitive to a number of other inputs (see Sec 1.2 for a more detailed discussion). Turbines are modeled in NWP simulations with wind farm parameterizations (WFPs, for a review see Fischereit et al., 2021), such as the Fitch WFP (Fitch et al., 2012), the Explicit Wake Parametrisation (EWP, Volker et al., 2015), and the Abkar WFP (Abkar and Porté-Agel, 2015). WFPs modify mesoscale flow in two manners. First, all WFPs impart drag, thereby slowing winds



downstream of a turbine. Second, WFPs modify downwind turbulent kinetic energy (TKE), either through an exclusively implicit approach (imparted drag changes the shear profile, thereby generating TKE generation, as is done in the EWP) or additionally through an explicit approach (explicitly adding TKE at wind turbines, as is done in the Fitch WFP). A number of recent studies have explored the impact of explicitly added TKE on wakes (e.g., Fitch et al., 2012; Vanderwende et al., 2016; Siedersleben et al., 2020a; Tomaszewski and Lundquist, 2020; Archer et al., 2020) and, as such, we also examine this factor in this study.

In this paper, we address the question: How sensitive are modeled mesoscale wakes to the choice of PBL parameterization? Specifically, we compare Fitch WFP simulations with both MYNN (Nakanishi and Niino, 2009) and the recently developed NCAR 3DPBL (Kosović et al., 2020b; ?). Aside from simply quantifying differing wake effects, we also explore why differences occur. As such, we conduct two sets of numerical experiments. First, to study wake sensitivity in a context where it is relatively straightforward to isolate wake effects, we simulate wakes in pseudo-steady idealized environments under stable, neutral, and unstable conditions. We also examine the role of explicitly added TKE in this set of simulations. Second, we ground this analysis in the real world by simulating a month-long case study centered on planned wind plants off the U.S. east coast. In Section 2, we describe the two PBL schemes, the integration of the NCAR 3DPBL with the Fitch WFP in the WRF code, and the setup of the simulations. In Section 3, we discuss the results of the idealized simulations. In Section 4, we analyze the real simulations. In Section 5, we conclude and present broader takeaways. Before we discuss the analysis methodology, we first highlight several WFP sensitivity studies and summarize their quantitative findings.

## 1.2 Previous Sensitivity Studies

In our following sensitivity analysis, we will focus on time-averaged differences in wind speeds, TKE, and capacity factors between MYNN and the NCAR 3DPBL. To give context to the magnitudes of sensitivity demonstrated here, we highlight previous WFP sensitivity studies that quantify WFP effects in a similar manner to our analysis.

Fitch et al. (2012) simulated a 10-by-10 grid of 5-MW turbines in an idealized neutral environment. They examined mean vertical profiles of wind speed deficits within the plant using the default Fitch WFP configuration as well as a configuration where the Fitch WFP does not add any explicit TKE. They found that maximum wind speed deficits were about 0.4 m s$^{-1}$ weaker in the default configuration than in the configuration without TKE addition. They also tested the wind speed deficit sensitivity to vertical and horizontal resolution. Halving the vertical levels from the base-case of 81 levels showed "little sensitivity," especially within the rotor disk where wind speed deficits disagreed by less than ±0.1 m s$^{-1}$. Doubling the horizontal resolution to 2 km decreased the wind speed deficit within the rotor disk by about 0.1 m s$^{-1}$ and changed TKE shifts in the vicinity of the plant by ±0.1 m$^2$ s$^{-2}$.

Lee and Lundquist (2017) simulated four days of the CWEX-13 campaign in Iowa. They reported that mean 120-m winds in turbine-free simulations varied by 0.1–0.4 m s$^{-1}$ when they switched between a near-surface vertical grid resolution of 12 m and 22 m. These winds varied by 0.1–0.3 m s$^{-1}$ when the forcing product was switched between ERA-Interim and the Global Forecasting System. The study also compared observed and modeled total power production from a plant with 200 1.5-MW





turbines, finding that the bias shifted between 4 MW and 10 MW when vertical grid resolution was varied and 1 MW and 7
90  MW when the forcing product was varied.

Mangara et al. (2019) simulated three weeks of onshore winds in flat terrain in Changyi, China. They examined the impact
of horizontal and vertical resolution on modeled wakes. They calculated relative wind speed deficit in the most-downwind
grid cells that contained turbines, and they found that 19-day averaged hub-height relative wind speed deficits ranged between
approximately 4% and 10% when horizontal resolution was varied and approximately 5% and 9% when vertical resolution
95  was varied. They also found that added averaged TKE values could significantly vary with grid resolution, finding that added
hub-height TKE in the last row of the plant varied between approximately 0.15–0.50 $m^2\ s^{-2}$, depending on the grid resolution.

Shepherd et al. (2020) conducted a yearlong simulation over Iowa with the Fitch WFP and the EWP. They calculated
vertical profiles of median velocity deficits inside the Pomeroy wind plant cluster, finding that Fitch could predict winds that
were 1 $m\ s^{-1}$ stronger than in the EWP. The two schemes also predicted different heights for maximum wind speed deficit,
100  where the EWP produced the strongest deficits within the rotor disk, whereas, at times, Fitch produced wake maxima that lofted
above the rotor disk. External wakes (here defined as the distance at which wind speed deficits at all heights were smaller than
0.2 $m\ s^{-1}$) have a similar length for both PBL schemes, although wind speed deficits inside the external wake were stronger
with Fitch because of its stronger internal waking. The wind speed deficits correspondingly produce different power losses,
where the EWP predicts 5% more total annual power production than Fitch. Finally, due to the fundamentally different manner
105  in which TKE is treated within the WFPs, median values of TKE can be 1 $m^2\ s^{-2}$ stronger with Fitch than with the EWP inside
the plant.

Pryor et al. (2020) simulated 9 months of winds over Iowa with a number of model configurations. When they increased the
number of vertical levels from 41 to 57, monthly mean hub-height wind speeds increased between 0.2 $m\ s^{-1}$ and 0.6 $m\ s^{-1}$
in turbine-free simulations. However, when horizontal grid resolution was doubled from 4 km to 2 km in simulations with
110  41 vertical levels, the turbine-free hub-height wind speed distributions were statistically similar. The differences in maximum
hub-height TKE enhancement from the Fitch WFP in each month of the study varied between 0.2 $m^2\ s^{-2}$ and 0.3 $m^2\ s^{-2}$,
depending on the grid resolution. Median gross capacity factors simulated with the Fitch WFP were 38%–41%, depending on
the resolution, and they were similarly 41%–45% with the EWP.

Siedersleben et al. (2020b) conducted a WFP sensitivity study and compared their model data with observations from one
115  day of flights over offshore wind plants in the North Sea as part of the WIPAFF project. In the sensitivity study, they varied
horizontal grid resolution, vertical grid resolution, advection of TKE within the MYNN PBL scheme, presence of explicit TKE
addition, and the magnitude of the thrust coefficient. In the end, instantaneous wind speeds above the plant differed by as much
as 1.5 $m\ s^{-1}$ from the base case, depending on the combination of sensitivity factors. Similarly, instantaneous TKE varied by
more than 1 $m^2\ s^{-2}$ at a given location in space. The study also concluded that "the most important ingredient" for realistic
120  offshore wind plant simulations is the ability of WRF to accurately simulate the state of the atmosphere, irrespective of the
WFP configuration.

In summary, the 6 highlighted works quantified the sensitivity to horizontal grid resolution, vertical grid resolution, TKE
addition, and the choice of WFP. It is difficult to succinctly summarize the wind speed and TKE sensitivity to each of these





factors because of differences in the analysis approach (e.g., height of quantified wind differences, instantaneous versus aver-
aged winds). However, these studies often observe wind speed shifts of approximately 0.1 m s$^{-1}$ to 0.5 m s$^{-1}$. When two WFP
simulations that have explicit TKE turned on are compared, typical TKE shifts are approximately 0.1 m$^2$ s$^{-2}$ to 0.3 m$^2$ s$^{-2}$ .
When a WFP simulation with explicit TKE addition turned on is compared to one without explicit TKE addition, typical TKE
shifts are in the vicinity of 1 m$^2$ s$^{-2}$.

## 2 Methods

### 2.1 MYNN and the NCAR 3DPBL

The simulations in this paper are carried out using WRF v4.3.0 with two PBL schemes: MYNN (Nakanishi and Niino, 2009;
Olson, 2019) and the NCAR 3DPBL (Kosović et al., 2020a; **?**). The WRF v4.3.0 code was modified to include the NCAR
3DPBL code, which is being prepared for public release. For simplicity, we refer to the NCAR 3DPBL as simply "the 3DPBL."
Both MYNN and the 3DPBL share a common origin—they are fundamentally rooted in the turbulence modeling of Mellor
and Yamada (1974). Here, we use the level 2.5 MYNN and 3DPBL schemes, which both treat TKE as a prognostic variable,
thus improving their utility for wind turbine modeling, because generated TKE is advected by the PBL schemes.

These PBL schemes treat turbulent mixing differently. MYNN computes the vertical turbulent mixing by calculating the
vertical turbulent stress divergence, and it allows the horizontal turbulent mixing to be handled externally with a Smagorinsky-
type approach (Skamarock et al., 2021, Sec. 4.2). In contrast, the 3DPBL directly accounts for horizontal turbulent mixing by
explicitly computing the turbulent flux divergences for momentum, heat, and moisture. The 3DPBL has been implemented into
WRF to allow for three different configurations following the original Mellor-Yamada developments: (*i*) a full 3D model, (*ii*)
a quasi-3D model using the so-called PBL-approximation, and (*iii*) a 1D model using the PBL-approximation. In this analysis,
we employ the second option, as the full 3D parameterization is currently too computationally expensive for the month-
long mid-Atlantic simulations. When using the second option, the 3DPBL scheme handles both the vertical and horizontal
turbulent mixing by computing the 3D turbulent stress divergence, in addition to the 3D turbulent flux divergence of heat and
moisture. The vertical turbulent fluxes in the 3DPBL are calculated similarly to MYNN, and the horizontal turbulent fluxes
are calculated analytically following Mellor and Yamada (1982) after applying the PBL approximation (i.e., neglecting the
horizontal derivatives of mean quantities in addition to the vertical derivative of vertical velocity).

Aside from different approaches for horizontal mixing, the two PBL schemes also employ different master length scales and
closure constants. Both schemes employ one "master" length scale, although they calculate them differently. In the simulations
in this study, the 3DPBL master length scale follows Mellor and Yamada (1982), whereas the MYNN master length scale uses
a different approach that simultaneously accounts for length scales that characterize buoyancy, the surface layer, and the PBL
depth. The closure constants for the 3DPBL length scale come from Mellor and Yamada (1982), whereas the MYNN closure
constants were updated in Nakanishi and Niino (2009). The MYNN updates focused on convective conditions and, as such, we
expect (and find) that the two PBL schemes to behave most differently in convective conditions.





While the values of empirical constants are different, MYNN and the quasi-3DPBL use the same formulation to parameterize turbulent momentum, heat, and moisture fluxes. For example, they parameterize the vertical flux of the $u$-component of wind speed as

$$\langle uw \rangle = -LqS_m \frac{\partial U}{\partial z},$$ (1)

where $L$ is the master length scale, $q$ is $\sqrt{2\,\text{TKE}}$, $S_m$ is a stability function, and $U$ is zonal velocity (Mellor and Yamada, 1982).

The two PBL schemes also use identical formulations for components of the TKE budget. We briefly recount their formulations here, because we discuss TKE budgets in Sec 3.6. Precise details of each formulation, as well as definitions of each variable, can be found in Nakanishi and Niino (2009), Olson (2019), and **?**. Both PBL schemes calculate shear-generated TKE

as $P_s = -\left(\langle uw \rangle \frac{\partial U}{\partial z} + \langle vw \rangle \frac{\partial V}{\partial z}\right)$, buoyancy-generated TKE as $P_b = -\beta g_i \langle w\theta_v \rangle$, the diffusion of TKE as $\frac{\partial}{\partial x_k} LqS_q \frac{\partial q^2}{\partial x_k}$, and TKE dissipation as $\varepsilon = q^3/B_1 L$. While the formulations are identical, the budget components take on different values in each PBL scheme due to the value of different empirical constants and the numerical implementation of each calculation.

## 2.2    Integration of the Fitch WFP with the 3DPBL

To simulate wakes with the 3DPBL, we first integrated the Fitch WFP with the 3DPBL inside the WRF code. The Fitch WFP

modifies flow in two key manners (Fitch et al., 2012; Archer et al., 2020): by slowing winds

$$\frac{\partial u_k}{\partial t} = -\frac{1}{2} \frac{A_k C_T U_k u_k}{z_{k+1} - z_k}$$ (2)

$$\frac{\partial v_k}{\partial t} = -\frac{1}{2} \frac{A_k C_T U_k v_k}{z_{k+1} - z_k}$$ (3)

and by adding TKE

$$\frac{\partial \text{TKE}_k}{\partial t} = \frac{1}{2} \frac{A_k c_a C_{TKE} U_k^3}{z_{k+1} - z_k}.$$ (4)

In the above equations, $k$ is the vertical level that intersects the rotor, $A_k$ is the area of the rotor on this vertical level, $C_T$ is the thrust coefficient, $U_k$ is the wind speed, $u_k$ is the zonal wind, $v_k$ is the meridional wind, and $z_k$ is the height. The turbulence coefficient $C_{TKE}$ is calculated as the difference between the thrust coefficient $C_T$ and the power coefficient $C_P$. The thrust and power coefficients are functions of wind speed that are unique to a particular wind turbine, and their values are specified in the input file *wind-turbine.tbl*. The coefficient $c_a$ was introduced by Archer et al. (2020) to empirically modify the amount

of explicit TKE addition and, in this study, we either set it to 0 or 1.





| Stability | Label | Short Description |
|---|---|---|
| Stable | NWF | No-wind-farm simulation, forced by 10 m s$^{-1}$ winds and -15 W m$^{-2}$ surface cooling |
| | 100TKE | Turbine-including simulation with 100% of explicit TKE addition from the Fitch WFP and same forcing as NWF |
| | 0TKE | Turbine-including simulation with 0% of explicit TKE addition from the Fitch WFP and same forcing as NWF |
| Neutral | NWF | No-wind-farm simulation, forced by 10 m s$^{-1}$ winds and 0 W m$^{-2}$ surface cooling |
| | 100TKE | Turbine-including simulation with 100% of explicit TKE addition from the Fitch WFP and same forcing as NWF |
| | 0TKE | Turbine-including simulation with 0% of explicit TKE addition from the Fitch WFP and same forcing as NWF |
| Unstable | NWF | No-wind-farm simulation, forced by 10 m s$^{-1}$ winds and 20 W m$^{-2}$ surface heating |
| | 100TKE | Turbine-including simulation with 100% of explicit TKE addition from the Fitch WFP and same forcing as NWF |
| | 0TKE | Turbine-including simulation with 0% of explicit TKE addition from the Fitch WFP and same forcing as NWF |

**Table 1.** A summary of the different idealized simulation cases. Each of these cases was run with both MYNN and the 3DPBL.

The major challenge in integrating the Fitch WFP and the 3DPBL is that the 3DPBL code is housed in the dynamics (*dyn_em/*) part of the code, as opposed to the physics (*phys/*) part of the code where most other PBL schemes reside. As such, the codebase was modified to account for the user-selected PBL scheme. A call to the Fitch WFP's *dragforce* subroutine was added to the end of *dyn_em/module_first_rk_step_part2.F*. When called for the 3DPBL, the velocity tendencies and
TKE tendencies are additionally scaled by the column-mass in order to match the identical scaling that happens to the *phys/*-calculated tendencies earlier within *dyn_em/module_first_rk_step_part2.F*. Additionally, whereas the Fitch WFP code modifies the MYNN TKE field directly (including a timestep factor of $\partial t$), the new code modifies the 3DPBL TKE tendency field (omitting a factor of $\partial t$ and letting the rest of the code carry out the time integration).

### 2.3 Configuration of Idealized Simulations

First, we carry out a series of idealized simulations to study the effect of the PBL scheme on simulated wake dynamics in a simple flat terrain environment. All simulation inputs can be found on Zenodo (https://doi.org/10.5281/zenodo.5565399). We use the neutral idealized simulations of Fitch et al. (2012) as inspiration for our simulations, but we make a number of modifications. All simulations use two domains, each 202-by-202 grid points in the horizontal. MYNN is always used in the outer domain, whereas the inner domain is either MYNN or the 3DPBL. The outer domain uses a horizontal grid spacing of 3
km and a timestep of 9 seconds, whereas the inner domain uses a horizontal grid spacing of 1 km and a timestep of 3 seconds. The vertical grid uses 81 cells, up to a height of 20 km. Vertical grid stretching is employed to provide finer resolution near the surface, thereby allowing 28 vertical levels below a height of 300 m, following the recommendation of Tomaszewski and Lundquist (2020) for nominally 10 m of resolution near the surface. All simulations have a roughness length of 0.1 m. All simulations are forced with 10 m s$^{-1}$ geostrophic winds, thereby situating the turbines within Region II of the power curve,
where the turbine thrust coefficient is largest and power production is a function of wind speed cubed. Stable simulations are





additionally forced with -15 W m$^2$ surface cooling, and unstable simulations are forced with 20 W m$^2$ surface heating. These sensible heat flux values were chosen based on typical simulated conditions at Vineyard Wind (Rosencrans et al., in prep), and are smaller than typical values over land. Simulations for each stability case are initialized with a neutral temperature profile of 285 K within the boundary layer up to 500 m. The boundary layer is capped with a two-layer inversion: a strong inversion (5 K

warming between 500 m and 600 m) and a weaker inversion (3 K km$^{-1}$ lapse rate above 600 m). One turbine-free simulation is spun up for three days for each stability and each PBL scheme, after which, pseduo-steady conditions are achieved. This long spinup is done to dampen inertial oscillations, as in Fitch et al. (2012). After spin up, the boundary-layer height is approximately 600 m in the stable simulations, 750 m in the neutral simulations, and 1,150 m in the unstable simulations.

After spinning up turbine-free simulations, we run three cases of simulations for each of the stabilities and each of the PBL

schemes (Table 1). The first case is simply a continuation of the turbine-free simulations and is referred to as the no-wind-farm (NWF) case. The second case (100TKE) includes a 10-by-10 grid of 12-MW International Energy Agency (IEA, Beiter et al., 2020) reference offshore wind turbines placed in the center of the inner domain. The turbine hub-height is 138 m, and the rotor diameter is 215 m. Cut-in speed is 3 m s$^{-1}$, rated speed is 10.9 m s$^{-1}$, and cut-out speed is 30 m s$^{-1}$. Turbines are placed 2 km apart, which is close to the 1 nautical mile spacing used in the real simulations. In this case, 100% of explicit TKE is generated

by the Fitch WFP ($c_a = 1$). In the third case (0TKE), we explore the sensitivity to explicitly added TKE by duplicating the setup of the second case, but turning off explicit TKE generation ($c_a = 0$).

### 2.4   Configuration of Real Simulations

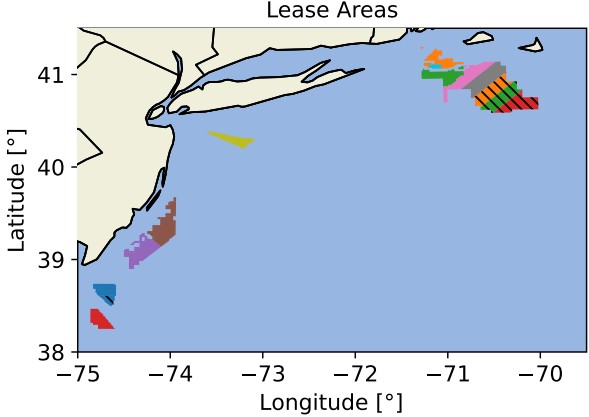

**Figure 1.** The offshore Lease Areas simulated in the LEASE case (colored and possibly hatched). Vineyard Wind 1 (gray) is simulated as the lone farm in the VW-ONLY case.

Next, to ground this analysis in a real-world application, we study wake sensitivity centered on planned offshore wind plants off the U.S. east coast (Fig. 1). The domain is centered on the mid-Atlantic, focused on waters off the coast of Maryland,

Delaware, New Jersey, New York, Connecticut, Rhode Island, and Massachusetts. We run three cases of real simulations for





| Label | Short Description |
|---|---|
| NWF | No-wind-farm simulation |
| LEASE | A simulation that contains turbines in all discussed Lease Areas |
| VW-ONLY | A simulation that contains turbines only at Vineyard Wind 1 |

**Table 2.** A summary of different real simulation cases. Each of these cases was run with both MYNN and the 3DPBL.

both of the PBL schemes (Table 2). The first case simulates a turbine-free atmosphere and is referred to as the NWF case. The second case simulates turbines in all of the lease areas within the domain as defined by the Bureau of Offshore Energy Management (BOEM) on March 3, 2020 (BOEM, 2020). As such, 14 lease areas are included, ranging from US Wind Inc. in the south to the cluster of lease areas near Rhode Island and Massachusetts in the north. This case is called LEASE. The third
case is the same as the second case, but it only simulates the Vineyard Wind 1 wind plant; it is referred to as VW-ONLY. This third case enables us to differentiate between two types of wakes, as proposed by the International Electrotechnical Commission (IEC) 61400-15 working group (Fields and Sherwin, 2017):

1. Internal wakes: These are wake effects that come from within a plant. The wakes at Vineyard Wind 1 in VW-ONLY only come from one farm. As such, we can isolate self-waking in VW-ONLY.

2. External wakes: These are wake effects that arrive from outside of a particular plant of interest. The wakes at Vineyard Wind 1 in LEASE come from the Vineyard Wind farm itself as well as the neighboring farms. We can subtract out the internal waking from the VW-ONLY effects to quantify the impact of the neighbors.

We focus on Vineyard Wind because, at the time the simulations for this study were initiated, it was likely to be first 100+ MW project in US offshore water.
We simulate winds for the month of August 2020. We chose this period because of its high electricity demand (Livingston and Lundquist, 2020). We start the simulations on July 30, 2020, to allow for 48 hours of spin-up, and we omit this data from analysis. The domain used in this study is identical to the one used by the 20-year NREL mid-Atlantic analysis available at NREL (2020) as well as Rosencrans et al. (in prep). The outer domain at 6-km grid spacing has 196 grid points in the west-east direction and 122 grid points in the south-north direction and uses a timestep of 18 seconds. The inner domain of 2-km grid
spacing has 466 grid points in the west-east direction and 259 in the south-north direction and uses a timestep of 6 seconds. We save data from the inner domain every 5 minutes. As with the idealized simulations, the outer domain for all simulations uses the MYNN PBL scheme, whereas the inner domain either uses MYNN or the 3DPBL. TKE advection is turned on for all MYNN domains. All domains use Thompson microphysics (Thompson et al., 2008), RRTMG for radiation (Iacono et al., 2008), Jiménez-modified Monin-Obukhov for the surface layer scheme (Jiménez et al., 2012), and the unified Noah land-
surface model (Tewari et al., 2004). Horizontal turbulent mixing is carried out with a Smagorinsky-diffusion style approach in





MYNN, whereas it is calculated from the horizontal turbulent flux divergence in the 3DPBL. ERA5 (Hersbach et al., 2020) provides atmospheric forcing, and OSTIA at 1/10° resolution provides sea surface temperatures (Donlon et al., 2012).

We select turbines and turbine spacing that are consistent with current expected standards of offshore U.S. wind plants. As was done in the idealized simulations, we use the 12-MW IEA reference turbine in the real simulations. This is similar to
the 62 13-MW turbines that are slated for operation in Vineyard Wind 1 (Vineyard Wind, 2021). All modeled lease areas are fully built out in order to study the maximum possible power production and wake strength. The turbine spacing is identical to that used by Rosencrans et al. (in prep). They are spaced 1 nautical mile apart, which is the same spacing that will be used in Vineyard Wind 1. In total, 177 turbines are modeled in the VW-ONLY simulations, and 1,418 turbines are modeled in the LEASE simulations. We note that the official layout of the Vineyard Wind 1 site was announced after our simulations were
completed. The official layout will only include turbines in the northern half of Vineyard Wind 1. As such, our study will overestimate the magnitude of internal waking at Vineyard Wind 1 in the years immediately after it is built.

## 3 Results: Idealized Simulations

### 3.1 Turbine-Free Conditions

Before comparing the impacts of the turbines on the wind flow, it is necessary to characterize the baseline wind speeds and
TKE in the turbine-free simulations. All simulations are forced with 10 m s$^{-1}$ geostrophic winds, but wind speed profiles are additionally shaped by the varying heat fluxes. Under stable conditions, MYNN and the 3DPBL produce similar winds across the rotor disk in the NWF simulations (Fig. 2a,b). Under stable conditions, MYNN hub-height wind speeds are 7.55 m s$^{-1}$, and 3DPBL hub-height wind speeds are 7.38 m s$^{-1}$. Under stable conditions, MYNN and 3DPBL wind speed profiles exhibit a large degree of shear across the rotor disk. Stable MYNN hub-height wind speeds are 0.17 m s$^{-1}$ (2%) stronger than they are
in the 3DPBL under stable conditions. Both PBL schemes produce a low-level jet under stable conditions, and the "nose" of the jet is stronger in the 3DPBL above the rotor disk. Under neutral conditions, MYNN and the 3DPBL also produce similar wind speed profiles. MYNN hub-height wind speeds are 8.04 m s$^{-1}$, and 3DPBL hub-height wind speeds are 7.82 m s$^{-1}$. Here, MYNN hub-height wind speeds are 0.22 m s$^{-1}$ (3%) stronger than in the 3DPBL.

In these neutral and stable conditions, the two PBL schemes can produce different amounts of TKE in NWF simulations
(Fig. 2d,e). Under neutral conditions, the two PBL schemes produce similar levels of TKE at all heights, approximately 0.65 m$^2$ s$^{-2}$ at the bottom of the disk and 0.40 m$^2$ s$^{-2}$ at the top of the disk. TKE linearly decays in both schemes from its maximum value at the surface to approximately zero above the capping inversion. MYNN produces slightly more TKE within the disk, whereas it produces slightly less TKE above. Under stable conditions, the two PBL schemes generate distinct TKE profiles. MYNN shows a linear decay of TKE across the disk. This decay is quicker than it is under neutral conditions. In contrast, the
3DPBL has a near-constant TKE value of 0.45 m$^2$ s$^{-2}$ within the disk that tapers off to 0 m$^2$ s$^{-2}$ as the atmospheric boundary layer (ABL) height is approached.

Of the three stability conditions, the greatest NWF wind speed and TKE differences arise under unstable conditions (Fig. 2c,f). MYNN produces substantially faster wind speed profiles across the rotor disk, and its hub-height wind speeds (9.62 m

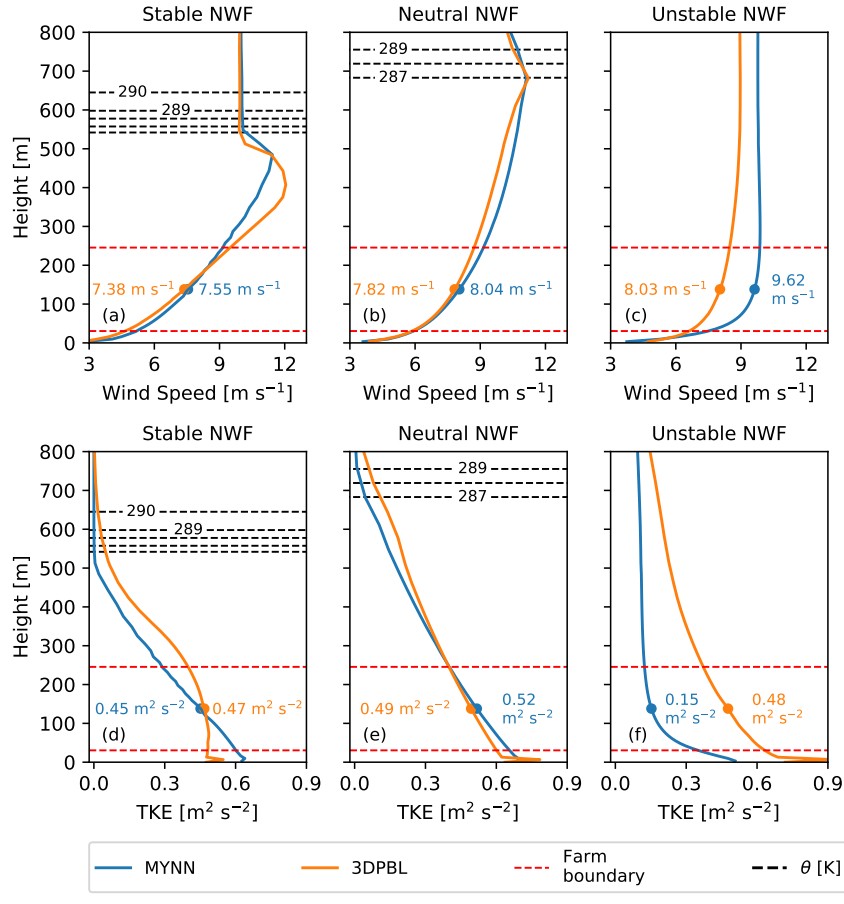

**Figure 2.** Horizontally averaged wind speed profiles (a–c) and TKE profiles (d–f) in different stabilities. Hub-height values of the fields in each PBL scheme are noted in each panel. The black dashed lines denote potential temperature contours and convey a sense of the boundary layer depth.

s$^{-1}$) are approximately 1.5 m s$^{-1}$ stronger than in the 3DPBL (8.03 m s$^{-1}$). MYNN wind speeds approach the 10 m s$^{-1}$

geostrophic wind speed at a height of 100 m above ground, whereas the 3DPBL winds remain below 9 m s$^{-1}$ within the first 800 m above ground. The 3DPBL produces approximately 2–3 times as much TKE than MYNN throughout the rotor disk. For example, MYNN hub-height TKE is 0.15 m$^2$ s$^{-2}$, whereas 3DPBL hub-height TKE is 0.48 m$^2$ s$^{-2}$. These large differences under unstable NWF conditions likely arise in part from different empirical constants and length scale formulations within the two PBL schemes (Sec. 2.1).

The differences and similarities in the wind profiles of the NWF simulations will dictate the comparison of the wake effects between the PBL schemes in turbine-including simulations. Differences in wake magnitude will largely be governed by differences in the wind speeds entering the plant as differences in wake recovery processes, which are linked to parameterizations of turbulent fluxes (Gupta and Baidya Roy, 2021). The turbine thrust coefficient is nearly identical for all simulations based





on hub-height wind speeds, 0.803–0.806, simplifying the comparison. Additionally, the two PBL schemes produce similar
NWF wind speed profiles under neutral and stable conditions. Thus, in these two conditions, we expect that wake differences
will primarily arise from differences in the way that turbulent fluxes are modeled within each PBL scheme. Under unstable
conditions, however, the differences in NWF wind speeds will additionally play a major role in wake dynamics. Finally, we
note that NWF hub-height wind speeds increase in strength when moving from stable to neutral to unstable conditions. This
wind speed variability will impact power production and losses when comparing between stability conditions.

## 3.2  Hub-Height Wind Speed Deficits

Internal wakes are sensitive to the choice of PBL scheme and stability (Fig. 3). We quantify internal wake strength by finding
the average hub-height wind speeds within the plant in the turbine-including simulations ("WFP," which is a generic stand-in for
"100TKE" or "0TKE") relative to hub-height winds in the turbine-free simulations ("NWF"). We also calculate the percentage
of wind speed loss with reference to the NWF winds inside the plant, and we calculate percentage point [pp] differences as
differences between percentage values. The idealized MYNN simulations consistently show internal wakes that are stronger
than their 3DPBL counterparts at hub-height. These differences primarily arise from two sources—differences in inflow wind
speed and differences in turbulent mixing formulations. The neutral and stable simulations have similar inflow wind speeds.
As such, for neutral and stable cases, the differences in internal wakes are more driven by differing wake recovery rates than
by differing inflow wind speed. Recovery rates differ between PBL schemes because different empirical constants are used to
calculate turbulent fluxes and because the PBL schemes have different ambient vertical profiles of TKE (particularly in stable
conditions). Internal wakes in the stable and neutral MYNN simulations are 0.20–0.27 m s$^{-1}$ (2.4 pp–3.5 pp) stronger than they
are in the 3DPBL. By contrast, the wakes under unstable conditions show larger absolute discrepancies because hub-height
MYNN inflow wind speeds are about 1.5 m s$^{-1}$ stronger than they are in the 3DPBL (Fig. 2c). Unstable MYNN internal wakes
are 0.31–0.34 m s$^{-1}$ (1.5 pp–2.0 pp) stronger than they are in the 3DPBL simulations.

Internal wakes are also sensitive to the amount of TKE explicitly added by the Fitch wind farm parameterization. The
addition of TKE modifies turbulent fluxes and, therefore, turbulent mixing and momentum recovery (Eq. 1). When explicit
TKE addition is turned off (Fig. 3d–f, j–l), hub-height wind speed deficits inside the plant consistently amplify by 0.09–0.17 m
s$^{-1}$ (1.0 pp–2.3 pp). Thus, an increase in TKE across the rotor disk increases hub-height wake recovery, likely due to increased
mixing with strong winds aloft (Sec. 3.3). The increase in deficits does not show a clear correlation with PBL scheme. In
some stabilities (stable, neutral), MYNN shows greater sensitivity to TKE, whereas the 3DPBL shows greater sensitivity in
other stabilities (unstable). Similarly, the increase in deficits does not show a clear correlation with stability. For example, the
strongest shift in MYNN wind speed deficit occurs under stable conditions, but the strongest 3DPBL shift occurs under neutral
conditions. Additionally, we note that in these idealized simulations, internal wakes are typically more sensitive to PBL scheme
choice than the presence or absence of explicitly added TKE.

While external wakes are more nebulous to characterize, they also show sensitivity to atmospheric stability. There is no
singular standard approach that is used to characterize external wakes (Fischereit et al., 2021), so we adopt two approaches: by
identifying the contours of the 0.5 m s$^{-1}$ deficit as well as the contours of the $e$-folding distance. MYNN internal wakes are

**Figure 3.** Hub-height wind speed deficits in varying stabilities (left-right) and PBL configurations (up-down). Average hub-height wind speed deficits inside the plant are noted—both in absolute magnitude as well as a percentage relative to the NWF winds.





all substantially stronger than 0.5 m s$^{-1}$, and as such, the 0.5 m s$^{-1}$ deficit contour can be used to directly compare MYNN simulations to one another. As expected (Fitch et al., 2013; Porté-Agel et al., 2020), the 0.5 m s$^{-1}$ contours extend furthest

under stable conditions, more than 80 km in the 100TKE MYNN simulation, and 70 km in the 0TKE simulation. Surprisingly, the 0.5 m s$^{-1}$ contours extend slightly further (2 km–4 km) in the unstable simulations than they do in the neutral simulations. While extensive observations and other simulations suggest that wakes erode faster in convective conditions, we note that the wake erosion is caused by ambient TKE. The behavior here occurs both because hub-height unstable NWF MYNN winds are approximately 1.5 m s$^{-1}$ stronger than they are under neutral conditions, and the unstable simulations actually have only half

as much hub-height TKE than the neutral simulations (Fig. 2f). Larger amounts of downwind TKE encourage wake recovery (an effect which is observed in these simulations and will be discussed momentarily). Thus, the larger TKE under neutral MYNN also helps erode the neutral MYNN wakes in these simulations.

We additionally characterize external wakes in the idealized simulations by using the $e$-folding distance. Internal wakes in the 3DPBL have values closer to 0.5 m s$^{-1}$; thus, the 0.5 m s$^{-1}$ deficit contour is less useful to characterize external

wakes. Instead, we calculate the $e$-folding contour as $1/e$ times the average internal wake strength, or about 36% (Fitch et al., 2012). This metric characterizes external wakes relative to internal wake magnitude. In the 100TKE 3DPBL simulations, the stable and neutral $e$-folding distance extends beyond 80 km, whereas it is 25 km in the unstable simulations. Thus, the unstable 100TKE 3DPBL simulations show quicker wake recovery, as expected. By contrast, the $e$-folding distance in the 0TKE 3DPBL simulations show little sensitivity to stability, as all the wake lengths are 12 km–17 km, perhaps due to similar values of ambient

turbulence in the 3DPBL simulations (Fig. 2d-f).

Depending on the presence of TKE addition and stability, MYNN and the 3DPBL can produce similar external wakes. The addition of explicit TKE inside the plant consistently either has negligible impact on external wakes or it extends the external wakes further. This is somewhat counterintuitive on first glance, as the addition of explicit TKE weakens internal wakes. However, the external wake length growth may be associated with a slight decrease in turbulent mixing due to a small

decrease in TKE downwind of the plant (as observed in Fig. 6g and Fitch et al. (2012)). Under unstable conditions, the addition of TKE had a relatively small impact on the 0.5 m s$^{-1}$ deficit contour and $e$-folding distance under unstable conditions for both MYNN and the 3DPBL. Under neutral and stable conditions, the addition of TKE extended the $e$-folding distance by dozens of kilometers for both PBL schemes. Under these two stabilities, external wakes show greater sensitivity to TKE in the 3DPBL than in MYNN. Thus, based on this relative $e$-folding metric, the addition of TKE appears to extend the external wake.

Additionally, the addition of TKE extends the 0.5 m s$^{-1}$ deficit contour in the stable MYNN simulations. However, it does not change the location of the same contour in the neutral MYNN simulations.

The imperfect description of external wakes complicates their direct comparison between the two PBL schemes. At times, the $e$-folding distance is longer in MYNN simulations, but at other times, it is longer in the 3DPBL. For example, the $e$-folding distance extends to 70 km in the stable 0TKE MYNN simulation, whereas it is only 17 km in the stable 0TKE 3DPBL

simulation. By contrast, the $e$-folding distance is 45 km in the neutral 100TKE MYNN simulations but more than 80 km in the 100TKE 3DPBL simulations. However, we see relatively small changes when comparing the unstable simulations. Thus, in the end, external wakes under neutral and stable conditions can substantially differ depending on PBL scheme, but one



PBL scheme does not consistently predict a stronger wake. Both schemes produce relatively similar wakes under unstable conditions.

We briefly digress from the discussion on wakes to note that upwind blockage (Schneemann et al., 2021; Sanchez Gomez et al., 2021) can be observed in these idealized simulations and tangential flow accelerations, similar to the speed-ups seen by Nygaard and Hansen (2016), can be observed adjacent to the wakes. Blockage is strongest in the stable conditions, where 0.125–0.250 m s$^{-1}$ deficits extend 5 km–7 km upwind of the plant. Under neutral conditions, blockage of the same magnitude extends 3 km–5 km upwind of the plant. Blockage does not appear in the unstable simulations. In general, blockage here is

a function of stability but not PBL scheme or TKE addition. The hub-height wind acceleration neighboring the wakes is also a function of stability (strongest in stable conditions, weakest in unstable conditions), but it also varies with TKE addition (stronger acceleration when TKE addition in turned on).

### 3.3  Vertical Structure of Wind Speed Deficits

While hub-height winds are particularly important to quantify, it is also helpful to characterize wakes over the vertical extent

of the rotor disk. We calculate the wind speed deficit averaged along the $y$-extent (predominantly crosswind) of the plant for each simulation (Fig. 4).

Just as MYNN consistently predicted stronger hub-height wind speed deficits, it also tends to predict stronger internal wakes over the extent of the rotor disk. MYNN tends to predict maximum wind speed deficits inside the plant that are 0.25–0.50 m s$^{-1}$ stronger than those of the 3DPBL. The one exception to this is the stable 100TKE simulations (Fig. 4a,g), in which the

3DPBL produces stronger wind speed deficits (by 0.50 m s$^{-1}$) in the upper half of the disk. The stable 100TKE 3DPBL wind speed deficits (Fig. 4g) are particularly strong above the rotor disk (1.70 m s$^{-1}$), where the low-level jet and NWF wind speeds are fastest. In the neutral and stable simulations, the 3DPBL produces stronger wind speed deficits than MYNN above the rotor disk, whereas the two PBL schemes produce similar magnitude deficits there under unstable conditions.

Vertical profiles of wind speed deficits are sensitive to the presence or absence of explicit TKE addition. Under stable and

neutral simulations (e.g., Fig. 4a,d), the addition of TKE drives the location of the maximum deficit upward by 50 m–100 m. In the case of the stable 3DPBL, the addition of TKE also significantly increases the magnitude of the maximum wind speed deficit (by more than 1 m s$^{-1}$). The addition of TKE only marginally changes the magnitude of the maximum deficit in the remaining stable and neutral simulations. TKE addition also promotes flow acceleration below hub height in these stability conditions. Acceleration under the rotor disk was observed in Bodini et al. (in review) but not in Archer et al. (2019). Wind

speed deficits under unstable simulations are relatively insensitive to the addition of TKE. When TKE addition is turned off, the location and magnitude of maximum wind speed deficits only marginally shift in the unstable simulations.

### 3.4  Difference in Momentum Tendencies

In large part, the two PBL schemes produce different wind speed deficits in their wakes because the schemes parameterize turbulent fluxes differently, as we illustrate here. The $u$ and $v$ components of wind speed are modified by mechanisms such as

advection of the mean wind, the Coriolis force, and the divergence of the turbulent momentum fluxes (Stull, 1988, Eqn. 5.1a).

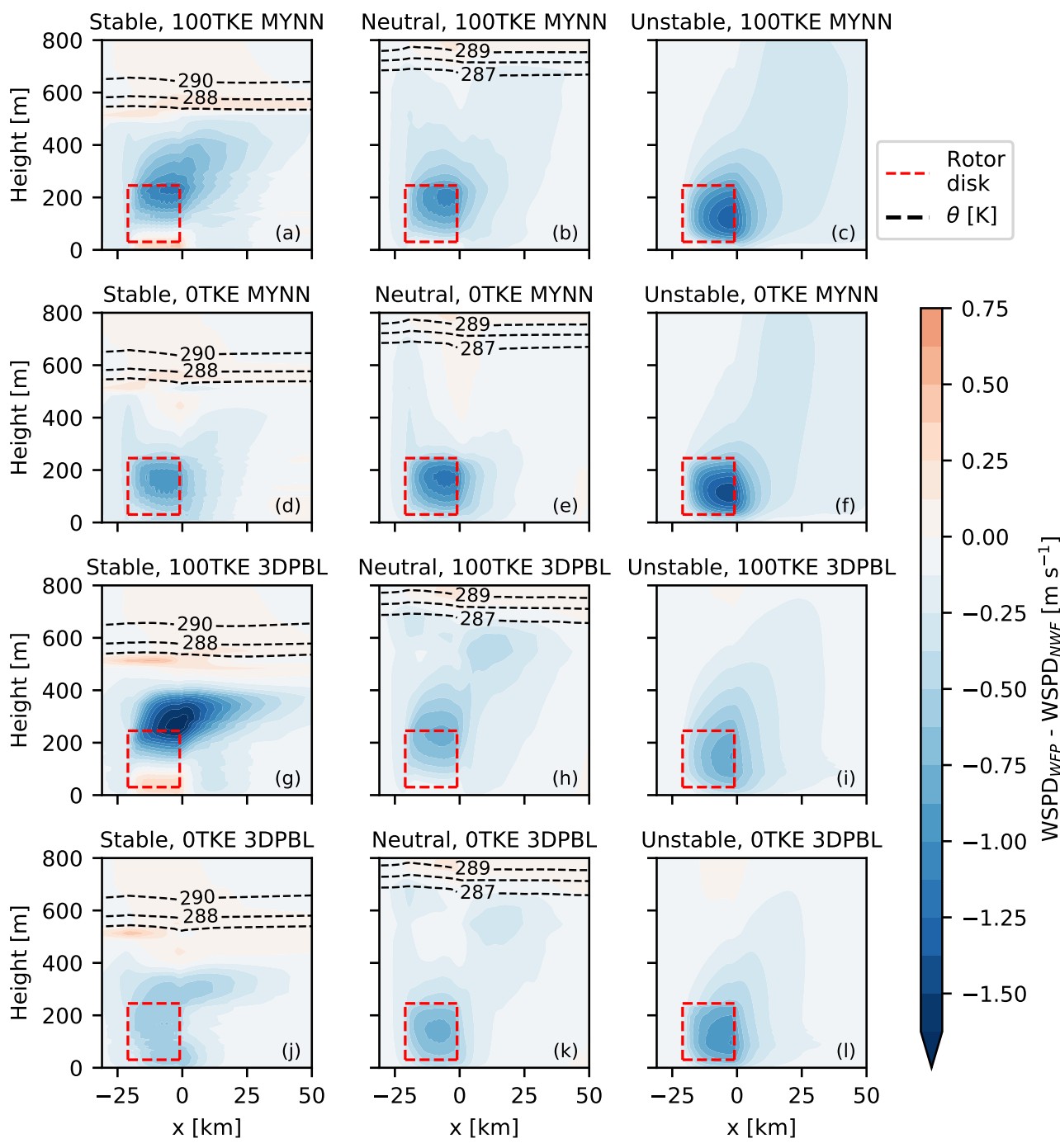

**Figure 4.** Side view of horizontally averaged wind speed deficits in varying stability conditions (left-right) and PBL configurations (up-down). The height of the ABL is visualized in the stable and neutral simulations with $\theta$ contours.



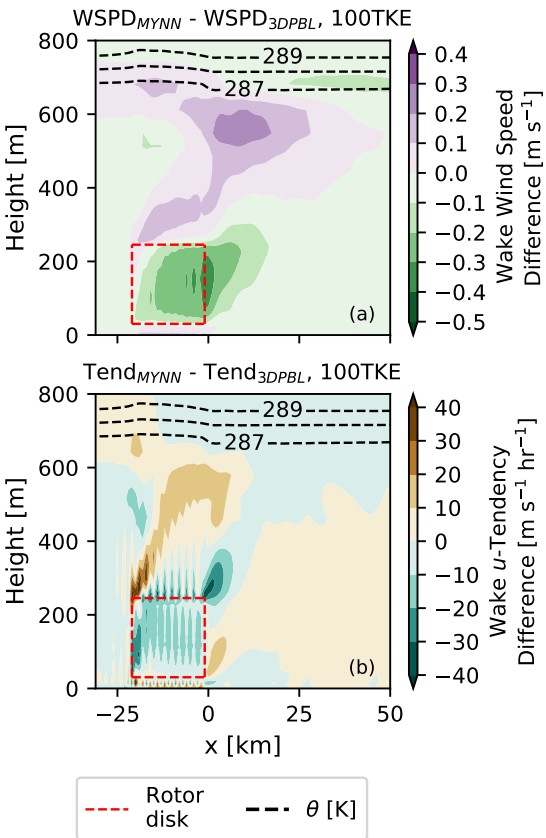

**Figure 5.** (a) Side view of the difference in wind speed deficits in the neutral 100TKE simulations. For example, Figure 5a was calculated as panel Figs. 4b–4h. (b) The $u$-tendency deficits in the 100TKE are calculated in a similar manner. Potential temperature $\theta$ values that have been averaged over the $y$-extent of the plant are taken from the MYNN simulations.

We expect all these terms, aside from the divergence of turbulent momentum fluxes, to be similar for both MYNN and the 3DPBL, as the NWF wind speeds are similar but two PBL schemes parameterize momentum fluxes uniquely. We calculate the $u$-tendency due to the turbulent flux divergence as the vertical derivative of $\overline{u'w'}$, neglecting the horizontal components of flux divergence because they are significantly smaller in the 3DPBL than $\overline{u'w'}$, and they are not computed in the MYNN parameterization. We also omit visualizations of $v$-tendency because they are substantially smaller than the $u$-tendency in these idealized simulations forced with a $u$ geostrophic wind. We investigate the relationship of wind speeds and turbulent fluxes between the two PBL schemes in the neutral 100TKE simulations by comparing two fields in the wakes—the wind speed deficits and the turbulent flux divergence $u$-tendency "deficits" (Fig. 5). The $u$-tendency deficits are defined as tendencies in the turbine-free simulations subtracted from tendencies in the turbine-including simulations.

The differences in tendency deficits between the two PBL schemes drive the differences in the wind speed wakes. As winds advect primarily along the $x$-direction, wind speed magnitudes are modified by the tendency. For example, the $u$-tendency





is more negative for MYNN in the first few columns of the plant. Correspondingly, the MYNN wind speed deficits are more negative. They become more negative as the winds pass over regions where MYNN continues to have a more negative tendency, and they become more positive as they pass over regions where MYNN has a more positive tendency. Thus, the modeled wind

speed deficits in the wake of a plant depend on how the PBL scheme parameterizes turbulent momentum fluxes.

## 3.5  Total TKE

While wind speed deficits are often significantly sensitive to PBL scheme and stability, shifts in TKE have a more variable response to these factors (Fig. 6). When the Fitch WFP is turned on, changes in TKE are constrained within the horizontal extent of the plant and tend to not advect far downwind. This stands in contrast to the real onshore WRF WFP simulations

of Mangara et al. (2019), which saw substantial TKE changes 20 km–30 km downwind. In the idealized simulations with explicit TKE addition turned on, maximum TKE shifts are typically between 0.5 $m^2$ $s^{-2}$ and 0.7 $m^2$ $s^{-2}$. TKE is strongest above hub height, as in the turbine-resolving large-eddy simulations of (Vanderwende et al., 2016) as well as the WRF WFP simulations of Mangara et al. (2019). The TKE within the rotor disk of the 100TKE MYNN simulations does not change much with stability. The 3DPBL simulations are more sensitive to stability changes, particularly in the stable simulations, where

maximum increases in TKE approach 0.8 $m^2$ $s^{-2}$ at the top of the rotor disk. This is the only 100TKE simulation that shows significant TKE accumulation in one region, as TKE is relatively horizontally homogeneous inside the plant throughout the rest of the 100TKE simulations. The stable 100TKE 3DPBL simulation is also the only simulation to show a significant decrease in TKE (0.1–0.2 $m^2$ $s^{-2}$) relative to the NWF simulations downwind of the plant. This decrease in downwind TKE may be linked to the growth of external wake lengths (Sec. 3.2).

All 0TKE simulations tend to show minor TKE generation (0.1–0.3 $m^2$ $s^{-2}$) at the top of the rotor disk where shear is the strongest, and they also show small TKE deficits (0.1–0.2 $m^2$ $s^{-2}$) below hub height relative to the NWF simulations. The one exception is the unstable 0TKE MYNN simulation, which shows negligible TKE changes when turbines are included. Thus, while TKE magnitudes in these mesoscale simulations are sensitive to the amount of TKE explicitly added by the Fitch wind farm parameterization, they are relatively insensitive to shifts in atmospheric stability and PBL scheme. Additionally, we note

that TKE fields appear "smoother" in the 3DPBL simulations than they do in the MYNN simulations. This smoothing likely arises from differences in the way that TKE advection, empirical constants, and horizontal mixing are handled within each PBL scheme.

## 3.6  TKE Budget

The changes in total TKE fields fundamentally arise from differences in how different components of the TKE budget (Stull,

1988, Eqn. 5.1a) are parameterized. We calculate profiles of four budget components over the horizontal extent of the plant (Sec. 2.1): shear-generated TKE, buoyancy-generated TKE, vertical turbulent transport of TKE, and TKE dissipation. We visualize the value of the of the turbine-free profiles subtracted from the turbine-including profiles (Fig. 7).

In many scenarios, the two PBL schemes have similar TKE budgets, explaining the often similar total TKE fields in the vicinity of the plants. Buoyant production or destruction of TKE is near-zero across all simulations, as all simulations are



**Figure 6.** Side view of horizontally averaged TKE in varying stabilities (left-right) and PBL configurations (up-down). The height of the ABL is visualized in the stable and neutral simulations with $\theta$ contours.





**Figure 7.** Components of the TKE budget that have been horizontally averaged over the extent of the plant, organized by budget component: column 1 is shear-generated TKE, column 2 is buoyancy generation or destruction of TKE, column 3 is vertical turbulent transport and column 4 is TKE dissipation. Row 1 shows these components in stable conditions, row 2 shows neutral conditions, and row 3 shows unstable conditions. Each "100TKE" and "0TKE" profile is calculated with respect to the NWF profiles as WFP-NWF.





forced by a zero or near-zero heat flux. The 0TKE simulations for both PBL schemes have similar values of shear-generated TKE and TKE dissipation at nearly all heights, regardless of stability. While noisy, the 0TKE simulations of both PBL schemes also have similar characterizations of the vertical diffusion of TKE.

Larger differences between MYNN and the 3DPBL emerge in the 100TKE simulations, although these differences can still be quite small, again explaining similar total TKE fields in the vicinity of the plants. While shear-generated TKE is near-identical for both PBL schemes in the neutral and unstable simulations, the 3DPBL produces substantially more shear-generated TKE above the rotor disk than MYNN under stable conditions (Fig. 7a). This greater shear-generated TKE translates to overall larger values of TKE in this region for the 3DPBL (Fig. 6). While profiles of the vertical turbulent transport of TKE are noisy, both PBL schemes tend to have negative TKE turbulent transport in the upper half of the rotor disk and positive TKE turbulent transport in the lower half of the disk. The magnitude tends to be stronger in MYNN than in the 3DPBL, regardless of stability. The 100TKE simulations can also show larger differences in TKE dissipation, particularly in unstable conditions (Fig. 7j). TKE dissipation is parameterized in both PBL schemes with a dependency on $q^3$, and as such, small differences in total TKE can lead to large discrepancies between the two PBL schemes.

### 3.7 Power

Power production and losses due to internal waking change with PBL scheme (Fig. 8). We calculate the capacity factor for each turbine, the average capacity factor of the plant, and the average power deficit due to internal wakes with reference to the NWF hub-height wind speed. Capacity factor is defined as the ratio of actual power output relative to the maximum possible power output. Average power production throughout the plant is a function of NWF hub-height wind speed and wake recovery rate (which is governed by turbulent fluxes in each PBL scheme). Under neutral and stable conditions, the 3DPBL consistently predicts larger capacity factors than MYNN by 1 pp–2 pp. This behavior occurs because the two PBL schemes have similar inflow speeds, but MYNN has stronger hub-height internal waking (Sec. 3.2). The similar NWF winds in these stabilities (differing by $\sim$0.25 m s$^{-1}$) allow us to isolate the effect of differing wake recovering rates between PBL schemes. Capacity factor losses due to internal waking in these stabilities are 3 pp–5 pp weaker in the 3DPBL than in MYNN. The 3DPBL power losses range from 6.6 pp–10.8 pp, whereas the MYNN power losses range from 9.9 pp–15.7 pp. Thus, in neutral and stable conditions, the 3DPBL shows a quicker wake recovery rate inside the plant.

By contrast, unstable NWF MYNN hub-height wind speeds are about 1.5 m s$^{-1}$ stronger than they are in the 3DPBL. As was the case in the other two stabilities, MYNN still predicts stronger losses due to internal wakes. The unstable MYNN capacity factor losses (27.7 pp–29.6 pp) are substantially stronger than the unstable 3DPBL losses (10.4 pp–12.1 pp) because power output in Region II of the power curve is more sensitive at larger values of wind speed (due to the dependency on wind speed cubed). However, unlike in stable and neutral conditions, in unstable conditions MYNN has an overall higher average capacity factor, as its substantially stronger NWF winds offset its stronger wakes.

Paralleling the behavior of hub-height wind speed deficits, the inclusion of explicitly generated TKE consistently leads to smaller wake-induced power losses. The addition of TKE led to a reduction of hub-height wind speed deficits (Sec. 3.2).





**Figure 8.** Heat maps of capacity factor for each turbine, based on the turbine's position in the plant. The average capacity factor and internal wake strength are noted for each simulation. The unstable MYNN simulations use their own colorbar because their capacity factor range is substantially different than that of the other simulations.





As such, the losses in the 100TKE simulations are approximately 1 pp–2 pp smaller than they are in the 0TKE simulations, regardless of stability or PBL scheme.

In the end, these power calculations emphasize that power losses are a function of wind speeds coming into a plant, wake recovery rate, and TKE addition. We stress that these idealized simulations have been carried out in pseudo-steady conditions, and they occur for one set of geostrophic winds in one part of the power curve. To better predict the cumulative non-linear interactions of the effects of these parameters on losses at a real-world location, we simulate a month-long case study in the U.S. mid-Atlantic coast. The interplay between incoming hub-height wind speeds and wake recovery rates becomes much

muddier in a time-varying environment.

## 4   Results: Mid-Atlantic Case Study

In this section, we compare wind speed deficits in wakes produced with MYNN and the 3DPBL in August 2020 in the mid-Atlantic. As described in Sec. 2.4, we run three categories of simulations—NWF, Vineyard Wind 1 only, and all the lease areas. We pay special attention to Vineyard Wind 1 because our set of simulations allows us to differentiate between internal and

external waking at this site. Finding only minor TKE differences between MYNN and the 3DPBL in the idealized simulations, we omit TKE analysis at this site and focus instead on wind speeds and power production.

### 4.1   Turbine-Free Winds

Before analyzing wake effects in the mid-Atlantic, we first examine wind speeds in turbine-free simulations. Specifically, we calculate average profiles at the Vineyard Wind 1 centroid. We classify each 5-minute interval output as stable, neutral, or

unstable. While a number of metrics can be used to classify atmospheric stability (e.g., bulk Richardson number, Obukhov length), we use WRF-predicted surface heat fluxes at the Vineyard Wind 1 centroid to facilitate comparison to the idealized simulations. We define stable conditions as having a heat flux less than -5 W m$^{-2}$, unstable conditions as having a heat flux greater than 5 W m$^{-2}$, and neutral conditions as in between. We designate these thresholds so they overlap with the stability metrics used in the idealized simulations. We emphasize that offshore heat fluxes in this domain are significantly weaker than

typical heat fluxes observed on land. As such, while we refer to atmospheric states as "stable" and "unstable" in this offshore environment, these states can resemble the atmosphere under near-neutral stratification. MYNN and the 3DPBL spend a similar percentage of the month under stable conditions (35% and 33%, respectively), whereas MYNN shows more frequent neutral conditions than the 3DPBL (50% versus 40%) and, conversely, MYNN shows less frequent unstable conditions than the 3DPBL (15% versus 27%).

The average NWF wind speed profiles at Vineyard Wind 1 share characteristics with wind speed profiles in the idealized simulations (Fig. 9). Stable profiles at Vineyard Wind 1 have the fastest wind speeds, and (as was the case in the idealized simulations) MYNN has a faster average hub-height wind speed than the 3DPBL (12.19 m s$^{-1}$ versus 11.21 m s$^{-1}$, respectively). This ∼1 m s$^{-1}$ difference is stronger than the ∼0.25 m s$^{-1}$ difference in the stable idealized simulations. The unstable simulations have the second-fastest wind speeds, where MYNN hub-height wind speeds are 9.78 m s$^{-1}$ and 3DPBL hub-height





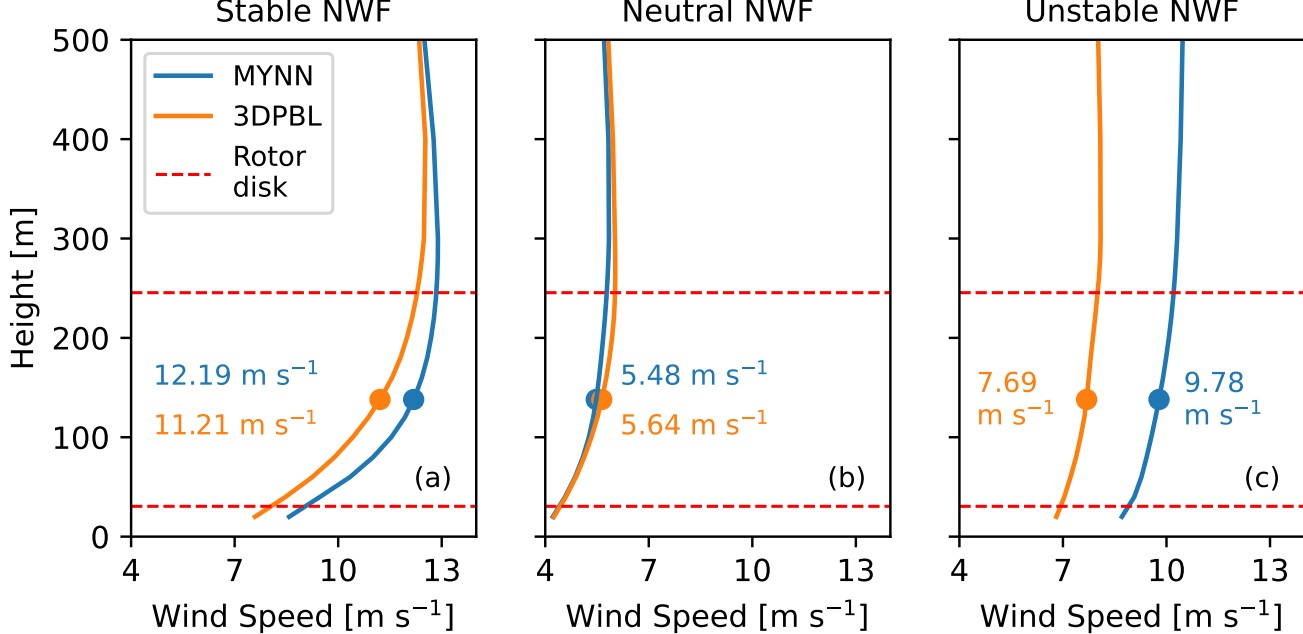

**Figure 9.** Averaged wind speed profiles at the Vineyard Wind 1 centroid for August 2020 simulations in (a) stable (surface heat flux < - 5 W $m^{-2}$), (b) neutral (-5 W $m^{-2}$ < surface heat flux < 5 W $m^{-2}$), and (c) unstable (surface heat flux > 5 W $m^{-2}$) conditions. Average hub-height wind speeds are noted.

wind speeds are 7.69 m $s^{-1}$. As was the case in the idealized simulations, in the real simulations MYNN produces substantially faster unstable NWF winds than the 3DPBL. Neutral conditions produce the weakest wind speeds for both PBL schemes, as average MYNN hub-height wind speeds are 5.48 m $s^{-1}$ and 3DPBL hub-height wind speeds are 5.64 m $s^{-1}$. The neutral Vineyard Wind profiles are similar across PBL schemes, as was the case in the idealized simulations, but, notably, the 3DPBL now shows slightly faster hub-height wind speeds.

We also characterize the distribution of hub-height NWF wind directions and wind speeds using a wind rose at the Vineyard Wind centroid (Fig. 10). Under stable and neutral conditions, winds are predominantly out of the southwest, whereas under unstable conditions, they are out of the northeast and presumably influenced by land, Martha's Vineyard. Thus, the wakes under unstable conditions will point to the southwest, opposite to those under neutral and stable conditions, which will be directed to the northeast. In all three stability conditions, the distribution of wind directions is fairly narrow, which facilitates

the appearance of wakes in time-averaged visualizations. MYNN shows a greater prevalence of winds at rated power (Region III) under stable and unstable conditions. In neutral conditions, both MYNN and 3DPBL winds are predominantly in Region II; therefore, we expect less wind resource (larger wake effects on power production) in this stability condition, at least for the month of August.



**Figure 10.** Wind roses at the Vineyard Wind 1 centroid showing the distribution of wind speed (binned by power curve region) with wind direction.



## 4.2 Average Hub-Height Wind Speed Deficits

We calculate the average hub-height wind speed deficits by stability in the mid-Atlantic (Fig. 11). We time average the hub-height wind speeds in the NWF, VW-ONLY, and LEASE simulations, categorizing the stability of each 5-min period based off heat fluxes at the centroid of the NWF Vineyard Wind simulations and using the same timestamps for all three wind turbine cases for each PBL scheme. The distribution of wind direction is narrow in each stability and, as such, we do not additionally filter by wind direction in the wake analysis as might be done at a site with more variable wind directions.

Monthly averaged internal mid-Atlantic wakes, quantified at Vineyard Wind 1, are typically sensitive to both stability as well as PBL scheme (Fig. 11a–c,g–i). Average stable internal wakes are strongest (2.0–2.2 m s$^{-1}$), whereas they are weaker in the neutral and unstable simulations (0.9–1.2 m s$^{-1}$). MYNN produces stronger internal waking in the stable VW-ONLY simulation than the 3DPBL does (2.21 m s$^{-1}$ versus 1.97 m s$^{-1}$, or 18.1% versus 17.5%). Similarly, MYNN also produces stronger absolute internal losses in the unstable VW-ONLY simulations than the 3DPBL (1.20 m s$^{-1}$ versus 0.94 m s$^{-1}$,

12.3% for both PBL schemes). Interestingly, this pattern is flipped in the neutral simulations where MYNN produces weaker internal waking than the 3DPBL (-1.02 m s$^{-1}$ versus 1.06 m s$^{-1}$, or 18.5% versus 18.7%). We also note that even though average unstable hub-height NWF wind speeds are 2.0–4.3 m s$^{-1}$ stronger than their neutral counterparts, the average absolute unstable MYNN internal waking is only 0.1–0.2 m s$^{-1}$ stronger than under neutral conditions. Offshore wakes under unstable conditions in these real simulations erode much more quickly than they would when forced under neutral or stable stratification

under the same winds, even when the surface heating is weak. This effect can also be seen by looking at relative waking effects—relative neutral internal wakes are about 18.6%, whereas they are 12.3% in unstable conditions.

The different PBL schemes lead to different internal waking, although it is more difficult to characterize the exact mechanisms in the time-varying mid-Atlantic simulations than in the pseudo-steady-state idealized simulations. The differences in hub-height NWF wind speed play a key role. In the mid-Atlantic simulations, when MYNN has a stronger NWF hub-height

wind speed, it produces a stronger averaged internal wake (as compared with stable and unstable conditions). The NWF wind speed differences are smallest under neutral conditions in which the 3DPBL is only ~0.25 m s$^{-1}$ stronger than MYNN at hub-height. In these conditions, it is easier to observe the impact of different turbulent mixing formulations. In the idealized neutral simulations, MYNN had slightly stronger NWF wind speeds (coincidentally, ~0.25 m s$^{-1}$ stronger), and its wakes were also stronger (also by ~0.25 m s$^{-1}$). In the real simulations, 3DPBL NWF winds are slightly stronger, but the internal

wakes have a near-equal magnitude between the PBL schemes (differences <0.05 m s$^{-1}$). Thus, the 3DPBL's faster internal recovery rate observed in the idealized neutral simulations appears to persist through to the mid-Atlantic simulations as well. Aside from differences in NWF wind speeds and turbulent mixing approaches, the two PBL schemes also may lead to different wake magnitudes due to effects from time-varying winds as well as interactions with other physics parameterizations that were not activated in the idealized simulations.

Monthly averaged external wake propagation is particularly sensitive to stability and only weakly sensitive to the PBL scheme choice (Fig. 11d–f,j–l). We characterize external wake propagation in the mid-Atlantic by highlighting the 0.5 m s$^{-1}$ contour and neglecting the $e$-folding distance, because the 0.5 m s$^{-1}$ contour is a sufficiently useful metric when all internal



**Figure 11.** Hub-height wind speed deficits in varying stabilities (left-right) and PBL schemes (up-down) at Vineyard Wind 1. Rows 1 and 3 (a, b, c and g, h, i, respectively) visualize wind speed deficits relative to their respective NWF counterparts at Vineyard Wind 1 only, whereas rows 2 (d, e, f) and 4 (j, k, l) show deficits resulting from all neighboring lease areas. Average hub-height wind speed deficits inside Vineyard Wind 1 are noted. The wind speed deficit percentage is calculated with respect to the NWF Vineyard Wind 1 average wind speed. The line segment used to quantify external wake length is shown in panels (a) and (g).



wakes are substantially stronger than 0.5 m s$^{-1}$. As expected, external wakes propagate furthest under stable conditions. In the stable VW-ONLY cases, they propagate 51 km east of the easternmost point in Vineyard Wind with MYNN and 38 km east

with the 3DPBL (Fig. 11a, g). In the LEASE cases, these external wakes grow in size, although a characteristic length is more difficult to quantify due to their irregular shape. In neutral conditions, the 0.5 m s$^{-1}$ deficit remains close to the plant with both PBL schemes because of the weaker overall wind speeds. While the external wakes under unstable condition also remain fairly close to the plants (again, due to overall weaker wind speeds), they do propagate slightly further in the MYNN simulations (due to overall stronger wind speeds).

Additionally, we quantify the impact of external wakes on hub-height wind speed deficits by focusing on wind speed deficits inside Vineyard Wind 1. We calculate the monthly averaged external wake effect as the average internal wake magnitude in the VW-ONLY simulations subtracted from the perceived internal wake at Vineyard Wind in the LEASE simulations (e.g., Fig. 11a subtracted from Fig. 11d). Unsurprisingly, the largest external wake effects are observed under stable conditions. MYNN has a stronger external wake effect (0.78 m s$^{-1}$ or 6.4 pp) than the 3DPBL (0.56 m s$^{-1}$ or 5 pp) in these conditions. Interestingly,

under unstable conditions, the 3DPBL has a stronger wake effect even though its internal waking is weaker than MYNN's. The unstable 3DPBL external wake effect is 0.41 m s$^{-1}$ (5.3 pp), whereas it is 0.33 m s$^{-1}$ (3.4 pp) in MYNN. Under neutral conditions, the two PBL schemes produce only marginally different external wake effects (MYNN at 0.26 m s$^{-1}$ and 4.8 pp, 3DPBL at 0.24 m s$^{-1}$ and 4.4 pp), as with their marginally different internal wake strengths.

### 4.3 Impact on Power Production

We quantify power production at Vineyard Wind 1 in the VW-ONLY simulations for each of the stability conditions (Fig. 12a–c, g–i). We calculate monthly averaged capacity factors for each of the grid cells within the plant. While wakes are strongest under stable conditions, power production is also largest under stable conditions due to the faster wind speeds. Even though average internal wind speed deficits were about 0.25 m s$^{-1}$ larger in the stable MYNN scenario, MYNN has a higher capacity factor than the 3DPBL (64.3% versus 59.7%). Thus, as was observed in the unstable idealized simulations, the power gain from the

substantially stronger NWF wind speeds overcomes the power decrease from the marginally stronger wakes. Power production is smallest under neutral conditions where winds are weakest. The two PBL schemes predict similar capacity factors under neutral conditions—15.2% for MYNN and 16.3% for 3DPBL. The largest capacity factor discrepancy occurs under unstable conditions. MYNN's average unstable capacity factor (56.1%) is much larger than the 3DPBL's (38.2%), correlating with MYNN's stronger NWF winds. This large discrepancy stems from the differences in wind speed distributions (Fig. 10c, f).

Depending on the PBL scheme, power losses associated with internal wakes disagree on the order of 1–2 percentage points. We calculate the expected unwaked power production in the idealized simulations by convolving the time-varying hub-height wind speeds with the power curve, and we calculate internal wake power loss with reference to this value. Under stable conditions, MYNN predicts slightly stronger losses than the 3DPBL (26.2 pp loss versus 24.9 pp loss). Under unstable conditions, we see the same general behavior—average MYNN losses are 12.4 pp and average 3DPBL losses are 10.3 pp. Even though

winds were much weaker overall under neutral conditions, we also see an approximate 1-pp difference in power loss. This time, MYNN predicts a slightly weaker loss (9.4 pp loss versus 10.5 pp loss). Thus, neither PBL scheme produces consistently





**Figure 12.** The spatial distribution of average capacity factors at Vineyard Wind 1, binned by stability. Four colormaps are used due to the spread of values by stability and PBL scheme—one for stable simulations, one for neutral simulations, one for unstable MYNN simulations, and one for unstable 3DPBL simulations.





stronger internal waking across all stability conditions. The mid-Atlantic simulations also highlight that even though average capacity factors incorporating wake effects may be drastically different between PBL schemes (e.g., 56.1% for MYNN versus 38.2% for the 3DPBL), the internal wake losses can still be similar.

External wakes can amplify losses (Fig. 12d–f, j–l), particularly under stable conditions. When the neighboring plants are simulated, average capacity factors decrease by 8.1 pp in the stable MYNN scenario and 5.9 pp in the stable 3DPBL scenario. Under neutral conditions, external wakes further decrease capacity factors by 2.0 pp regardless of PBL scheme. External wakes in unstable conditions reduce MYNN's average capacity factor by 3.3 pp and the 3DPBL's average capacity factor by 3.6 pp. Overall, the different PBL schemes disagree on external wake power losses by 0.3 pp–2.2 pp across the different stabilities.

We also calculate the average August 2020 capacity factors at Vineyard Wind 1 and across all the lease areas irrespective of stability. At Vineyard Wind 1, in the VW-ONLY simulations, MYNN's month-long average capacity factor is 38.5%; the 3DPBL's average capacity factor is 36.3%. External waking into Vineyard Wind 1 in the LEASE simulations reduces the month-long capacity factor there by 4.3 pp to 34.2% for MYNN and by 3.8 pp to 32.6% for the 3DPBL. Therefore, MYNN predicts stronger losses due to external waking by 0.5 pp. The average capacity factor across all lease areas in the LEASE

simulations is 36.0% for MYNN and 33.2% for the 3DPBL. Thus, for all of these scenarios, the 3DPBL predicts capacity factors that are 1.6 pp–2.8 pp less than in MYNN—or about 4.7%–7.8% less total power generated.

We additionally visualize timeseries of wind speeds and power production to underscore the key role that NWF wind speed plays on power production (Fig. 13). We highlight NWF hub-height winds, capacity factor, and wake effects for one week of the simulation. In general, when NWF hub-height winds at the Vineyard Wind 1 centroid are stronger for a given PBL scheme

(Fig. 13a), that PBL scheme also produces more instantaneous power (Fig.13c). This amplification is particularly true when NWF wind speeds are in the vicinity of the rated wind speed (between Region II and Region III), as occurs on August 23. This pattern also persists when winds are weaker (as they are between August 24–25), although it is less prominent because the wakes are also weaker. Differences in wind speed are less important when NWF winds exceed the rated speed (as on August 26), as internal wakes in the VW-ONLY simulation do not reduce power output. Even then, external wakes can further weaken

winds inside Vineyard Wind and reduce power output, as they momentarily do on August 26. Regarding external wakes, even minor differences in NWF wind direction between PBL schemes ($< 5°$, Fig. 13b) can produce instantaneous external wake effects that differ by a dozen pp or greater (Fig. 13e), as occurs on August 23.

While faster wind speeds tend to lead to a larger capacity factor for a given PBL, faster wind speeds do not necessarily cause stronger internal or external waking. During the first red-highlighted period of Fig. 13, MYNN has faster NWF wind

speeds but weaker internal waking. This disparity could be explained by subtle differences in NWF wind direction, significant spatial variability of wind directions within Vineyard Wind 1 (rendering the single-point measurement of wind direction less informative), or differences in mixing for each PBL scheme, among other factors.

In the end, these timeseries figures show that the two PBL schemes can predict capacity factors that differ by dozen of pp in a given moment. Thus, when seeking to characterize power production uncertainty, it may be even more beneficial to vary

PBL schemes for short-term wind forecasts than for long-term wind resource assessments.



**Figure 13.** (a) A week of hub-height NWF wind speeds at the Vineyard Wind 1 centroid. (b) Wind directions at the same location. (c) Spatially averaged capacity factor at Vineyard Wind 1 in the VW-ONLY simulation. (d) Internal waking, as characterized by differences in spatially averaged capacity factors between the NWF and VW-ONLY simulations. (e) External waking, as characterized by differences in spatially averaged capacity factors between the VW-ONLY and LEASE simulations. Periods with significant external waking are highlighted in red in panels (b) and (e).





# 5    Conclusions

In this analysis, we studied the sensitivity of NWP-modeled mesoscale wakes to two PBL schemes: the widely-used MYNN and the recently introduced NCAR 3DPBL. While studies have showed that NWP-modeled wind resource in turbine-free simulations can significantly vary with PBL scheme, the same sensitivity has not yet been studied in simulations with explicitly

resolved turbines. We examined modeled wake sensitivity in two contexts. First, we simulated pseudo-steady idealized stable, neutral, and unstable environments that are forced by 10 m s$^{-1}$ geostrophic winds. In this context, we also examined wake sensitivity to the amount of explicitly added TKE from the Fitch wind farm parameterization. Second, we ran a month-long case study in the mid-Atlantic United States, centered on the Vineyard Wind 1 wind plant.

In the idealized simulations, average hub-height wind speed deficits inside the plant were sensitive to two factors that differed

between the PBL schemes: the no-wind-farm (NWF) wind speed at hub height and the wake recovery rate. MYNN consistently predicted faster NWF hub-height wind speeds than the 3DPBL, by about 0.25 m s$^{-1}$ in the stable and neutral simulations and by about 1.5 m s$^{-1}$ in the unstable simulations. The ideal stable and neutral simulations suggested that MYNN has a weaker hub-height wake recovery rate than the 3DPBL inside the plant, although it was difficult to quantify the exact effect due to the slightly differing inflow wind speeds. In the 100TKE simulations (which correspond to the default Fitch configuration),

MYNN predicted average hub-height wind speed deficits inside the plant that were 0.20–0.34 m s$^{-1}$ stronger than in the 3DPBL, depending on the stability. When explicit TKE addition was turned off, these deficits strengthened by about 0.1–0.2 m s$^{-1}$, depending on the stability and PBL scheme. We also examined wind speed deficits downwind of the plant, finding that the two PBL schemes produced different external wakes under stable and neutral conditions, but relatively similar external wakes under unstable conditions. Neither PBL scheme consistently produced longer external wakes. When explicit TKE from

the Fitch wind farm parameterization was turned off, external wakes either became shorter or their length did not significantly change.

In the end, either PBL scheme could lead to stronger total power production in the idealized plants, depending on the relative effects of NWF winds and wake recovery rates. In general, however, MYNN produced faster winds and stronger wakes. In the stable and neutral 100TKE simulations, the 3DPBL predicted average capacity factors that were about 1–2 percentage points

higher than with MYNN (or produced about 3%–6% more overall power). However, due to the substantially faster NWF winds under unstable conditions, MYNN predicted an average capacity factor that is about 13 percentage points higher than with the 3DPBL (or produced about 35% more overall power). While either PBL scheme could produce a higher capacity factor depending on stability, MYNN consistently had stronger losses due to internal wakes. MYNN capacity factor losses in the 100TKE simulations were 9.9–27.7 percentage points; they were 6.6–10.4 percentage points for the 3DPBL. When

explicit TKE addition was turned off, power losses due to internal wakes consistently increases by about 1–2 percentage points, independent of stability and PBL scheme.

Wind speed deficits in the real-world mid-Atlantic simulations also depended on the relative importance of NWF winds and wake recovery rates, although disentangling their relative importance in these time-varying fields became more difficult. In these simulations, under unstable conditions, MYNN again predicted substantially faster average NWF hub-height wind





speeds than the 3DPBL, by about 2 m s$^{-1}$. But, in addition, stable NWF MYNN winds were about 1 m s$^{-1}$ stronger than
they are in the 3DPBL. The two PBL schemes continued to have similar neutral NWF wind profiles, except the 3DPBL now
had slightly stronger (∼0.25 m s$^{-1}$) hub-height winds. Correspondingly, MYNN hub-height wind speed deficits at Vineyard
Wind 1 were stronger than 3DPBL under stable and unstable conditions but weaker under neutral conditions, reinforcing the
central importance of NWF wind speeds. Additionally, we found that external wakes from neighboring plants significantly
impact winds at Vineyard Wind 1, particularly under stable conditions, when they reduced average hub-height winds by an
additional 0.56–0.78 m s$^{-1}$.

Average capacity factors and power losses due to wakes directly correlated with wind speed deficits. MYNN predicted higher
capacity factors than the 3DPBL at Vineyard Wind 1 under stable and unstable conditions, but lower capacity factors under
neutral conditions. In simulations with only Vineyard Wind 1, the average stable capacity factors were 64.3% and 59.7% for
MYNN and the 3DPBL, respectively. Under unstable conditions, MYNN had a substantially higher average capacity factor
than the 3DPBL (56.1% versus 38.2%) due to the faster winds, despite wake effects. When averaged across all of August 2020,
the 3DPBL generated 5.7% less power than MYNN when just Vineyard Wind 1 turbines were simulated and 7.8% less power
when all turbines in the nearby lease areas were simulated.

While the two PBL schemes produced substantially different capacity factors, they predicted relatively similar power losses
due to internal wakes, differing by only 1–2 percentage points. Average internal losses at Vineyard Wind 1 were about 25.5
± 0.6 percentage points under stable conditions, 10 ± 0.5 percentage points under neutral conditions, and 11 ± 1 percentage
point under unstable conditions. Average external wakes showed slightly more variability between PBL schemes but, in the
end, they further reduced average capacity factors at Vineyard Wind 1 by about 6–8 percentage points under stable conditions,
2 percentage points under neutral conditions, and 3–4 percentage points under unstable conditions. While the two PBL schemes
lead to different monthly averaged assessments of power production and wake losses, these differences could be substantially
larger at a given moment. At times, instantaneous capacity factors differed by more than 20 percentage points between the two
PBL schemes, and internal wake effects, as well as external wake effects, also showed momentary discrepancies of a similar
magnitude. Thus, short-term forecasts of wind power production may be especially sensitive to PBL scheme.

In the end, this analysis suggests wind farm parameterizations are indeed sensitive to the choice of PBL scheme. In general
(but not always), MYNN predicts stronger wind speed deficits inside of a plant than the 3DPBL, and MYNN tends to predict
stronger power losses due to internal wakes. But, MYNN also tends to predict stronger plant inflow wind speeds. Because of
these competing effects (inflow wind speed vs wake dissipation), a generalization is not possible for which scheme consistently
predicts larger or smaller power outputs.

Due to this sensitivity, we recommend that future wind energy planning studies that examine mesoscale model sensitivity
consider varying the PBL scheme, along with other model inputs that have been established in literature, such as grid resolution,
magnitude of explicit TKE addition, and the choice of wind farm parameterization (Fischereit et al., 2021). While it is difficult
to judge the sensitivity of PBL scheme choice relative to most of the other inputs, we note that changing the PBL scheme had
a larger impact than turning explicit TKE generation on or off in the idealized simulations. While this analysis centered on





flat environments, we anticipate that the 3DPBL may produce even larger differences with MYNN in complex terrain, as each
PBL scheme behaves differently in heterogeneous environments.

Further, because the 3DPBL scheme is intended for mesoscale simulations approaching the "gray zone" where horizontal
grid resolution is ˜100-1000m, these 2-km resolution simulations are at the edge of where its representation of horizontal
turbulent eddies are necessary. While few simulations with the wind farm parameterization have delved into this grey zone,
it would be interesting to extend these simulations to finer resolution within the grey zone to determine optimal choices of
scheme. Such extensions should compare simulated wakes to observed wakes, the next step in this work.

*Code and data availability.* Namelists for all simulations, time-averaged idealized WRF data, WRF code modifications, and analysis note-
books to reproduce all figures can be found on Zenodo (https://doi.org/10.5281/zenodo.5565399). For convenience, much of the same ma-
terial has also been uploaded to GitHub (https://github.com/rybchuk/wfp-3dpbl-sensitivity). Data for the month-long 3DPBL NWF and
LEASE mid-Atlantic simulations can be found by looking inside of year-long simulation data stored on the AWS Registry of Open Data
at a link that will be updated during the peer review process. Data for the month-long 3DPBL VW-ONLY simulation as well the MYNN
simulations is stored on the University of Colorado's PetaLibrary and is available upon request.

*Author contributions.* **Alex Rybchuk**: Methodology, Investigation, Data Curation, Writing – original draft preparation, Writing – review
& editing; **Timothy W. Juliano**: Methodology, Software, Writing – review & editing; **Julie K. Lundquist**: Project conception, Funding
acquisition, Project administration, Supervision, Writing – review & editing, **David Rosencrans**: Methodology; **Nicola Bodini**: Project
administration, Writing – review & editing; **Mike Optis**: Funding acquisition and Project administration.

*Competing interests.* The authors declare no competing interest are present.

*Acknowledgements.* We would like to thank Branko Kosović and Pedro Jiménez Munoz for their developments on the NCAR 3DPBL and
sharing an early version of the code. We also thank the Pangeo community for their support with "big data" analysis (Odaka et al., 2020), as
well as the Python community at large for the development of libraries such as Matplotlib (Hunter, 2007), Numpy (Harris et al., 2020), Xarray
(Hoyer and Hamman, 2017), Dask (Rocklin, 2015), Zarr (Miles et al., 2021), Cartopy (Met Office, 2010), and Python-windrose (Roubeyrie
and Celles, 2018). Finally, we would also like to thank the manuscript editor and reviewers. This work was conducted with support from
the National Offshore Wind Research and Development Consortium under Agreement No. CRD-19-16351. A portion of the research was
performed using computational resources sponsored by the Department of Energy's Office of Energy Efficiency and Renewable Energy
and located at the National Renewable Energy Laboratory. This work utilized resources from the University of Colorado Boulder Research
Computing Group, which is supported by the National Science Foundation (awards ACI-1532235 and ACI-1532236), the University of
Colorado Boulder, and Colorado State University. JKL's effort was partially supported by the National Science Foundation under CAREER
grant AGS-1554055.



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
