# Peer review of "The Sensitivity of the Fitch Wind Farm Parameterization to a Three-Dimensional Planetary Boundary Layer Scheme"

_Wind Energy Science, 2021_

## Author Comment (AC1)

**Reviewer 1**

Dear authors,

Your paper tries to assess the sensitivity of the Fitch scheme in wake simulations by implementing it on a newly developed PBL scheme. I am certain that this manuscript is interesting and has some value but I am not sure whether in its actual state displays correctly this value, since I think the reader cannot gain much from all the numbers you reach with your simulations. So in the following, you find a list of major and minor/specific comments.

Dear Reviewer, we sincerely thank you for taking the time to read through the manuscript and offering feedback. We have substantially updated our manuscript following your suggestions, and we believe that the article is now much stronger and more interesting to read. Please find an itemized response below where our response is marked in red text.

Major comments:

1. My main comment is that the reader does not gain much when reading your work (this sounds harsh but I will try to explain). You compare results from Fitch using two PBL schemes. You do not conclude which PBL scheme is better. You do not conclude which PBL scheme is better with Fitch either. You basically simulate wakes and report the results and basically claim that since the results can be quite different then you suggest to use many different PBL schemes when simulating mesoscale wakes to get an idea of the uncertainty. This is of course valid but then what? One could then say you need to try all wind farm parametrization to get an idea of the uncertainty (and all PBL schemes available). I would think this would be a waste of resources. What I think it would have been nice to see here is some arguments/analysis in which given the differences between the wake simulations one could already say something about the ability of the Fitch scheme to model wakes. Or, better, that the authors have developed a methodology which could use the results of the simulations to gain knowledge about the accuracy of the Fitch scheme. All the quantitative results with regards to the differences under the idealized and the mid-Atlantic case are not really used to anything (I mean the actual numbers) and will not really help anybody to discern anything about the Fitch scheme and/or PBL schemes used. Maybe, one could also simply say: what should be the maximum differences in wind speeds and turbulence deficits when simulating wakes with different PBL schemes when using the same farm parametrization?

While your feedback is blunt, we have taken it to heart. As this is your main comment, we have emphasized our response here and broken it up into five parts.

1. The motivation for this manuscript could have been clearer and stronger.

Both reviewers noted that this manuscript was not engaging, so we have updated the introduction to more clearly highlight the important and time-sensitive problem that our research addresses (L25-49). Our primary motivation for conducting this analysis is as follows: offshore wind resource assessments are necessary for the rapidly developing offshore wind industry, but these resource assessments suffer from a lack of quality observations in most of the U.S. offshore wind resource areas. Thus, the offshore research agenda within the U.S. has explicitly solicited researchers to improve uncertainty quantification for offshore wind resource assessments. This call has come from academia (e.g., Archer et al. (2014), "Meteorology for Coastal/Offshore Wind Energy in the United States Recommendations and Research Needs for the Next 10 Years", which states that Research Need #2 is Uncertainty Quantification, especially in the form of ensemble simulations, which is what our research enables) as well as U.S. federal scientific agencies (e.g., Shaw et al. (2019) "Workshop of research needs for offshore wind resource characterization", in which the need for uncertainty quantification is stressed multiple times). Our research directly addresses the need to improve uncertainty quantification of offshore wind resource.

 2. Our article improves the capability for uncertainty quantification, **which is distinct from model validation,** but also crucial to ensure reliable numerical models.

As addressed above, there is an established need to quantify uncertainty in numerical models of wind resource, especially for regions that lack high quality observations. The National Research Council of the National Academies published a book that describes the importance of (and distinctions between) uncertainty quantification and validation, "Assessing the Reliability of Complex Models: Mathematical and Statistical Foundations of Verification, Validation, and Uncertainty Quantification". While validation answers the question of "How accurately does the model represent reality for the quantities of interest?", uncertainty quantification addresses "How do the various sources of error and uncertainty feed into the uncertainty in the model-based predictions of the quantities of interest?". These questions are interrelated but distinct and co-equal parts of the verification, validation, and uncertainty quantification (VVUQ) process. The book states that "the first UQ task is to quantify uncertainties in model inputs", and we do precisely this by developing and implementing new WRF code that allows users to vary the PBL scheme in a wind farm simulation. In our paper, we demonstrate that indeed there is a great deal of wind resource uncertainty that is associated with the choice in PBL scheme. Perhaps our

most simple and most important finding is that "the 3DPBL generates 4.7%-7.8% less power than MYNN" in the period and test case considered in our analysis. This finding, while technically specific to our analysis, could have serious implications for the financial viability of the offshore wind industry when similar analyses are conducted by interested stakeholders, and we demonstrate this finding without employing observations. (We point out that observations of wakes in US waters are not available given the lack of offshore deployment in US waters.)

We concede, as the book notes, that "VVUQ tasks are interrelated". However, our work (1) enhances the ability of researchers to conduct VVUQ studies for wind farm simulations in the future and (2) demonstrates **for the first time** that, indeed, future VVUQ studies should vary the PBL scheme. Both reviewers note that our paper already feels too long, and as such, it would not be possible to additionally conduct a thorough validation study that meets the best practices delineated in the book, such as

- "Principle: Validation assessments must take into account the uncertainties and errors in physical observations (measured data)."
- "Best practice: If possible, use a broad range of physical observation sources so that the accuracy of a model can be checked under different conditions and at multiple levels of integration."
- "Principle: Validation and prediction often involve specifying or calibrating model parameters."
- "Principle: The uncertainty in the prediction of a physical [quantity of interest] must be aggregated from uncertainties and errors introduced by many sources, including discrepancies in the mathematical model, numerical and code errors in the computational model, and uncertainties in model inputs and parameters."

We believe that a future article that carries out such a validation study and meets these principles would indeed be valuable, especially when conducted in conjunction with observations from upcoming field campaigns that include focused efforts to characterize mesoscale wake effects (e.g., AWAKEN/ARISE) and atmospheric phenomena in the U.S. offshore waters (e.g., WFIP3).

3. Now that our idealized simulations do not share large-scale forcing, it is increasingly important that we simulate a real case study with both MYNN and the 3DPBL that share large-scale forcing.

As discussed in greater detail below, following your feedback we have re-run all the idealized simulations so that their NWF hub-height wind speeds match. This effort necessitated tuning the large-scale forcing for each simulation. As such, the idealized simulations and real simulations highlight two distinct effects now. The idealized

simulations explore the question of "How does the unique momentum recovery parameterization of each PBL scheme affect wakes", and (as we discuss in greater detail in the manuscript) the real simulations ask "How do differing predictions of hub-height wind speed affect wakes?". Thus, the real simulations that we have run take on even greater importance now.

4. The scope of our article is consistent with the scope of other published WFP sensitivity studies, some of which have been published in *Wind Energy Science.*

- Bodini et al. (2021) published a WRF wind resource assessment study in *WES* (no WFP, but ultimately addressing the same fundamental question of uncertainty associated with a PBL) in the offshore US that did not compare to observations. This paper employed a 16-member WRF ensemble that was generated using in-built WRF capabilities.
- Pryor et al. (2021) published a WRF WFP sensitivity in onshore US in *JAMC* that also did not compare to observations.

Thus, our methodology is consistent with other academic publications that address the same fundamental question we investigate.

 5. Final, itemized response to additional concerns raised in this comment

Thus, with this in mind:

- "One could then say you need to try all wind farm parametrization to get an idea of the uncertainty (and all PBL schemes available). I would think this would be a waste of resources": This approach is exactly how Archer et al. (2014) suggest conducting uncertainty quantification, and correspondingly, how uncertainty quantification has been proposed along these lines in turbine-free wind resource assessments (e.g. Bodini et al. (2021)) and turbine-including assessments (e.g. Pryor et al. (2021)). Our work is the first to expand the uncertainty quantification capabilities to account for variance in PBL scheme in turbine-including simulations, a factor that has often been shown to be the largest source of uncertainty in turbine-free simulations (e.g. Optis et al. (2020)).
- "All the quantitative results with regards to the differences under the idealized and the mid-Atlantic case are not really used to anything (I mean the actual numbers) and will not really help anybody to discern anything about the Fitch scheme and/or PBL schemes used." Taking this feedback, we have substantially reduced the number of overly detailed quantitative comparisons throughout the manuscript. However, we have retained the most important quantitative differences, such as

statements similar to "the 3DPBL generates 4.7 %–7.8 % less power than MYNN in August 2020".

2. Your introduction is way too long. Particularly Section 1.2 is not really needed and it falls very much into kind of the same comment I just made about many numbers (from a number of previous wake works) without much meaning. The last paragraph of that subsection can be kept and will be sufficient. Also all the Fitch studies before Archer et al. (2020) are wrong (due to the bug in the model) and so they should not be mentioned. Lastly, what the introduction really lacks is why trying now Fitch with the 3DTKE PBL scheme? Would it be better? More realistic? Totally wrong? I guess the reader would tend to think that a 3D PBL scheme is better than a "2D" one such as MYNN

Regarding the length: We have cut Section 1.2 and have adapted its most important contents into the rest of the introduction.

Regarding the TKE advection bug: We have updated the manuscript to include the statement " We note that some Fitch WFP simulations, those with TKE advection turned on prior to Archer et al. (2020) were subject to a bug in the WRF code, and as such, the results from these studies should be interpreted with caution" (L81-83).

Regarding the discussing the 3DPBL in the intro: In an effort to keep the introduction short and engaging, we discuss the motivation for the 3DPBL in the Methods (L105-108). Crucially, we also emphasize that the NCAR 3DPBL parameterization is different than the 3DTKE parameterization. We have updated the manuscript to clarify this "To avoid confusion regarding nomenclature of new turbulence models, we note that the NCAR 3DPBL is different from the 3DTKE PBL scheme (Zhang et al., 2018)" (L101-103).

3. Second paragraph in Sect. 2.1: these lines should be complemented with the formulations so that the reader can get an idea of the advantages/extensions of the 3D TKE PBL.

Thank you for this suggestion. We wanted to do include the 3DPBL formulations in the initial draft of the manuscript, but we ultimately omitted copying the formulations from Juliano et al. (2022) to this paper, as the matrix equations in the original paper take up an entire page (Ref. Fig. 1). We include the reference to the paper with these formulations. Instead, we qualitatively and quantitatively discuss the major differences between the two PBL schemes in the manuscript (L109-133).

222

$$\tag{19}$$

Accepted for publication in *Monthly Weather Review*. DOI 10.1175/MWR-D-21-0164.1.

Response Figure 1. The page-long 3DPBL equations, as documented in Juliano et al (2022).

4. Text between lines 198 and 208: due to the use of a new PBL scheme, it would be interesting to see the development with time of the turbine-free simulations and find out why 3 days are indeed needed to develop the ABL and reach quasi-steady state (it just reads as a quite extremely long spin up period). By the way, you do not

mention (I think) the type of boundary conditions during spin up and wake simulations.

Thank you for the suggestion. We have added a section discussing the spin-up of the simulations at the start of Section 3.1 (L235-242).

[Figure]

Response Figure 2. Figure 2 in the updated manuscript. Hub-height wind speed at the center of each domain during spin-up in the idealized turbine-free simulations. The last 24 hours of each simulation is taken as the performance period for the NWF simulations.

The boundary conditions for the idealized simulations during spin-up can be found in the updated Table 1. We now explicitly state that the wake simulations use the same boundary conditions as the turbine free simulations (L184).

5. Idealized simulations: to compare fairly the Fitch scheme with the two PBL schemes, you should aim to get the same wind speed and direction at hub height. Therefore you should not use the same geostrophic wind for all simulations The problem is clear for the unstable simulation where it does not make sense at all to compare the wake results given the large differences in wind speed

Thank you for suggesting that we adjust our idealized simulations to match hub-height wind speeds. We reran all the idealized simulations with this constraint in mind. Your suggestion made it much easier to compare wake effects across all stabilities and PBL schemes. We believe that the idealized results are much clearer now.

As a result of the new simulations, the entirety of Section 3 has been updated. The key findings of the new idealized results can be quickly ascertained from the updated Conclusions (discussed immediately below).

6. Conclusions: the manuscript is already quite long and so such a long conclusion (which is not really concluding statements) is not needed. Can the reader get some nearly like bullet-points from your work? Also, and in relation to my comment 1, there are way too many sentences with a number of values that do not mean much if you do not have a reference or measurements. This is very clear between lines 657 and 674

We have truncated the conclusion, putting the key takeaways in a bullet list (L515-535). We have significantly reduced the amount of numerical values included in the Conclusion to focus instead on the main science messages, and we retained only the most important numbers.

Specific comments:

7. Line 4: "were only compatible with one PBL scheme" this is a general comment but I guess you mean the specific case of the WRF model, which is not mentioned at that point.

We now clarify that MYNN has been the only PBL scheme to work with Fitch "as of the Weather Research and Forecasting model v4.3.3" (L4)

8. Line 8: "internal" the reader does not know what you mean by internal so maybe drop the work and be specific in the abstract

We have replaced the phrase "internal wake" with "wind speed losses within the plant" in Line 8, and we have also amended the rest of the abstract accordingly.

9. Line 10: add "atmospheric" before "stability"

The updated abstract no longer uses the phrase "stability".

10. Line 27 "their impacts in numerical" I guess you mean "their impacts on atmospheric variables when implemented in numerical weather…" or so

The text in Line 27 has been entirely removed in the process of more strongly motivating this paper.

11. Line 31: you provide some low and high losses… what are the cases for this? I mean these are because of the size of wind farms?

Lee and Fields (2021) summarize that average total wake losses have been reported to be as low as 6.1% and as high at 40%. The 6.1% number comes from Mortensen et al. (2012) and the 40% number comes from Tindal (2009). It is difficult for us to say specifically why these differ, as both of these sources are conference presentations, and we do not have

access to them. We speculate that this spread occurs because of a number of reasons. As you suggest, one possible reason is the differing size of wind farms.

12. Line 57: maybe delete "generation"

We have deleted the surrounding section of text in order to make the introduction more concise, but thank you for catching that error.

13. Line 64: you have some ? signs when making a particular reference… this is not the first time

All the missing references have now been corrected, thank you.

14. Line 155: remove "to" before "behave"

We have removed "to".

15. Line 164: again a ? in a reference

All the missing references have now been corrected, thank you.

16. Line 165: not sure whether you define "Sq"

In an effort to keep the manuscript engaging and cut down on overly detailed comparisons, we have removed the Results section regarding TKE budget. As such, we have struck the Methods content where "Sq" was mistakenly not defined.

17. Line 180: why not using the value suggested in Archer et al. (2020)?

We remain with 100% TKE because Larsén and Fiscereit (2021) saw better performance with 100% TKE instead of 25% TKE, as we now note in L233.

18. Line 198: why not using the roughness of the sea? These parameterizations are mostly used offshore and it will be more straightforward to compare to the mid-Atlantic case

The new idealized simulations now use the roughness of the sea (0.0001 m) instead of the previous land roughness.

19. Line 199: the sentence kind of suggests that the hub height wind speed is very close to the geostrophic wind speed but that is not necessarily the case. So why 10 m/s?

Thank you for correcting our statement. Some of our updated idealized simulations are now run with 9 m/s geostrophic winds, so we have removed the corresponding text. We initially decided to force with 10 m/s geostrophic winds to be consistent with the idealized Fitch et al. (2012) study.

20. Figure 1. This figure can be changed to show the model domains and maybe an inset with a zoom of the vineyard wind 1 with the turbine arrangement would be nice

We have updated Figure 1 to show the model domains as well as a zoom in of Vineyard Wind 1. We have attached the updated figure below for convenience.

[Figure]

Response Figure 3. Figure 1 in the updated manuscript.

21. Line 238 and similar: all these references to manuscripts in preparation are not useful. Particularly the one at this line is not needed (also that in line 252)

We have now removed all references to manuscripts that are in preparation.

22. Sentences in lines 264-265 and 286 are redundant given each of the sentences before them

The redundant sentence in L264-265 has been removed. The statement in L286 has been updated to improve clarity.

23. Figure 2: do you say somewhere whether these profiles are instantaneous output at some time? Are they spatial averages over the whole domain? Are they time-averaged? Also the profiles should be somehow smoother; however they show some weird peaks, e.g., the highest wind speed of the stable MYNN or those below the lowest farm boundary in the stable TKE

Thank you for noting our omission. We have updated the Methods section to note that the performance period of the simulations required running "for 24 hours" (L184). We have also updated the caption for Figure 2 to clarify that these profiles are horizontally averaged over the extent of the domain as well as averaged for a day.

The wind speed profiles of the updated simulations appear to be smoother, but please let us know if this is still an issue. The "weird" TKE peaks at the surface of the 3DPBL simulations arises from the vertical staggering (or destaggering) of how TKE is represented within the 3DPBL versus the surface layer scheme. This issue is being addressed in upcoming versions of the 3DPBL, but (based on the similarity of TKE between MYNN and the 3DPBL) this mismatch does not appear to strongly affect TKE profiles. We now note "The sharp peaks in TKE at the lowest level of the 3DPBL simulations are tied to the staggered representation of TKE in the new PBL scheme, and future versions of the 3DPBL will correct this issue." (Figure 3 caption)

24. Also about the result in Fig 2 for the unstable TKE: why is MYNN 3 times lower than 3DTKE? You mention this is related to the empirical constants but the stable and neutral ones seem fine

The structural changes from MYJ (which is where the 3DPBL constants come from in this work) to MYNN focused on convective conditions (L126-127). Thus, we expected and indeed did see the strongest differences between the two PBL schemes in unstable conditions.

25. Line 300 maybe you can add after "values" whether these are from instantaneous values at a given time

We now clarify that these values come from "daylong time-averaged hub-height wind speeds" (L268).

26. Figure 3 and related results: why not aligning the wind with x so that when you make the cross (side) analyses the plots are easy to digest

We align our side views of the wake with the x-axis following Fitch et al. (2012). For example, see Figure 2 in that paper.

27. Line 327 "wakes erode" not sure how general is the erode term in wakes, could you replace it by recovery? I think you have different instances with this

The concept of wakes eroding appears throughout literature. For example "Wakes erode very quickly during unstable conditions" in Bodini et al. (2017) and "The results showed that under unstable conditions, the wake eroded rapidly" in Sun et al. (2020). We like this mental image as a complement to "wind speeds recovering", as we believe it paints a more vivid picture and helps keep the language more engaging.

28. Line 384 the ref. to your not published work can be changed. Some others have seen this as well

The reference has been published during the process of the review for this manuscript, and we have updated the reference accordingly.

    29. Line 432 delete one "of the"

We have corrected this typo, thank you.

    30. Line 480 Not really true as TKE was quite different for the unstable case

We have removed the sentence entirely.

    31. Line 491: I think you need to add "on land" after "near-neutral stratification"

We have clarified that we are referring to "onshore" atmosphere (L405), thank you.

    32. Line 491-494: a figure with the frequency distributions of these surface heat fluxes would be nice

We decided to omit the heat flux figure from the manuscript as we believe this information borders on the line of being overly detailed, but we include the figure in our response here for your reference.

[Figure]

Response Figure 4. Heat flux distribution at the Vineyard Wind centroid.

    33. Figure 9 how are the winds above the boundary layer? Are they close? Maybe an inset showing the full profile would be nice

Due to the large size (>10 TB) of the monthlong 5-minute resolution dataset, we were only able to retain data in the lowest 500 m, and thus we are unable to show winds above the boundary layer. This approach is consistent with NREL's other wind databases.

    34. Line 516 replace "off" by "on"

We have replaced "off" with "on". (L428)

35. Line 600 "amplification" I am not sure what you actually mean

Thank you for catching our ambiguous wording. We have changed "This amplification is particularly true" to "This power production difference is particularly true" (L490).

36. There are way too many avoidable references and also many references to your work (where any of the coauthors are involved)

We have cut down on many of the references by removing Section 1.2. Please let us know if there are any specific references that are unwarranted in the updated manuscript.

---

## Author Comment (AC2)

**Reviewer 2**

The paper inserts for the first time the Fitch wind farm parameterization in the newly developed 3DPBL scheme in the WRF model. This is important and innovative because, as far as I know, the Fitch parameterization has only been coupled with the MYNN PBL scheme. It is therefore valuable to see how it would work with a different PBL scheme. However, the paper consists of pages and pages and pages of detailed and rather pointless differences between the results obtained with the two schemes, first with idealized cases, then with a series of simulations of a few offshore wind energy areas in the US Northeast, leaving however no useful information on which would be better for which cases and why. I am afraid that, in order for the paper to be acceptable in its current format, too much additional work would be required (i.e., redo all idealized runs and simulate a different real farm), as discussed next.

An alternative would be to remove Section 4 entirely. Adding a real case would be valuable if it allowed the authors to validate the 3DPBL+Fitch coupling, but it has no value in this manuscript unfortunately, it just adds pages and pages of minutia and repetition.

Dear reviewer, thank you for taking the time to read the manuscript and provide feedback. We have updated the manuscript following the feedback from both reviewers, and we believe the new manuscript is significantly stronger. We have substantially cut down on the amount of "detailed and rather pointless differences". Following feedback from Reviewer 1, we have rerun and significantly revised the idealized simulations to (1) match hub-height wind speeds and (2) use a roughness length for offshore conditions. Please find our comments below in red.

Major points

1. Although some of the co-authors have access and/or have participated in field campaigns that have collected plenty of data on wind farm wakes, inland and offshore, and on observed wind farm power production (e.g., Siedersleben et al. 2020 just to mention one), no comparison against any type of observations is offered in this study. Why did the authors choose to simulate the Vineyard Wind and the other U.S. wind energy areas, for which no data are available yet, when so many other farms with data are available? At a very minimum, high-resolution simulations (like HRRR) could have been used for the wind speed profiles for August 2020 for Figure 9. But, better yet, a different farm with actual wake observations should have been simulated instead.

1. The motivation for this manuscript could have been clearer and stronger.

Both reviewers noted that this manuscript was not engaging, so we have updated the introduction to more clearly highlight the important and time-sensitive problem that our research addresses (L25-49). Our primary motivation for conducting this analysis is as follows: offshore wind resource assessments are necessary for the rapidly developing offshore wind industry, but these resource assessments suffer from a lack of quality observations across most of the U.S. Thus, the offshore research agenda within the U.S. has explicitly solicited researchers to improve uncertainty quantification for offshore wind resource assessments. This call has come from academia (e.g., Archer et al. (2014), "Meteorology for Coastal/Offshore Wind Energy in the United States Recommendations and Research Needs for the Next 10 Years", which states that Research Need #2 is Uncertainty Quantification, especially in the form of ensemble simulations, which is what our research enables) as well as U.S. federal scientific agencies (e.g., Shaw et al. (2019) "Workshop of research needs for offshore wind resource characterization", in which the need for uncertainty quantification is stressed time and time again). Our research directly addresses the need to improve uncertainty quantification of offshore wind resource.

 2. Our article improves the capability for uncertainty quantification, which is distinct from model validation, but also crucial to ensure reliable numerical models.

As addressed above, there is an established need to quantify uncertainty in numerical models of wind resource, especially for regions that lack high quality observations. The National Research Council of the National Academies put out a book that describes the importance of (and distinctions between) uncertainty quantification and validation, "Assessing the Reliability of Complex Models: Mathematical and Statistical Foundations of Verification, Validation, and Uncertainty Quantification". They state that validation answers the question of "How accurately does the model represent reality for the quantities of interest?" and uncertainty quantification addresses "How do the various sources of error and uncertainty feed into the uncertainty in the model-based predictions of the quantities of interest?". These are interrelated but distinct and co-equal parts of the verification, validation, and uncertainty quantification (VVUQ) process. The book states that "the first UQ task is to quantify uncertainties in model inputs", and we do precisely this by developing new WRF code that allows users to vary the PBL scheme in a wind farm simulation. In our paper, we demonstrate that indeed there is a great deal of wind resource uncertainty that is associated with the choice in PBL scheme. Perhaps our most simple and most important finding is that "the 3DPBL generates 4.7%-7.8% less power than MYNN in August 2020". This finding could have serious implications for the financial viability of the offshore wind industry, and we demonstrate this finding without employing observations.

We concede, as the book notes, that "VVUQ tasks are interrelated". However, our work (1) enhances the ability of researchers to conduct VVUQ studies for wind farm simulations in

the future and (2) demonstrates for the first time that, indeed, future VVUQ studies should vary the PBL scheme. Both reviewers note that our paper already feels too long, and as such, it would not be possible to additionally conduct a thorough validation study that meets the best practices delineated in the book, such as

- "Principle: Validation assessments must take into account the uncertainties and errors in physical observations (measured data)."
- "Best practice: If possible, use a broad range of physical observation sources so that the accuracy of a model can be checked under different conditions and at multiple levels of integration."
- "Principle: Validation and prediction often involve specifying or calibrating model parameters."
- "Principle: The uncertainty in the prediction of a physical [quantity of interest] must be aggregated from uncertainties and errors introduced by many sources, including discrepancies in the mathematical model, numerical and code errors in the computational model, and uncertainties in model inputs and parameters."

We believe that a future article that carries out such a validation study and meets these principles would indeed be valuable, especially when conducted in conjunction with observations from upcoming field campaigns that include focused efforts to characterize mesoscale wake effects (e.g. AWAKEN/ARISE).

3. Now that our idealized simulations do not share large-scale forcing, it is increasingly important that we simulate a real case study with both MYNN and the 3DPBL that share large-scale forcing.

As discussed in greater detail below, following feedback from Reviewer 1 we have re-run all the idealized simulations so that their NWF hub-height wind speeds match. This necessitated tuning the large-scale forcing for each simulation. As such, the idealized simulations and real simulations highlight two distinct effects now. The idealized simulations explore the question of "How does the unique momentum recovery parameterization of each PBL scheme affect wakes", and (as we discuss in greater detail in the manuscript) the real simulations ask "How do differing predictions of hub-height wind speed affect wakes?". Thus, the real simulations that we have run take on even greater importance now.

4. The scope of our article is consistent with the scope of other published WFP sensitivity studies, some of which have been published in *Wind Energy Science*.

- Bodini et al. (2021) published a WRF wind resource assessment study in *WES* (no WFP, but ultimately addressing the same fundamental question of uncertainty associated with a PBL) in the offshore US that did not compare to observations. This paper employed a 16-member WRF ensemble that was generated using in-built WRF capabilities.
- Pryor et al. (2021) published a WRF WFP sensitivity in onshore US in *JAMC* that did not compare to observations.

Thus, our methodology is consistent with other academic publications that address the same fundamental question we investigate.

 5. Itemized response to the concerns raised here

Thus, in summary:

- " Why did the authors choose to simulate the Vineyard Wind and the other U.S. wind energy areas, for which no data are available yet, when so many other farms with data are available? ": We conduct an uncertainty quantification study in this paper. Our work (1) enhances the ability of researchers to conduct VVUQ studies for wind farm simulations in the future and (2) demonstrates for the first time that, indeed, future VVUQ studies should vary the PBL scheme. We believe future validation studies will be important and complement the uncertainty quantification analysis undertaken here.

2. The authors state that TKE advection is turned on (see l. 243), but it does not seem to be true. Figure 6 shows without doubt that all the added TKE is confined within the boundaries of the wind farm and above it, but not advected downwind at all. With the MYNN scheme in particular (top two rows), one can even see the individual positions of the turbines, one every other grid cell, with the added TKE at their grid cells and above, but none in the next adjacent cells downwind. This proves that no advection is actually operating. The authors need to double check that bl_mynn_tkeadvect is indeed set to true in the inner domain. Since TKE advection appears to be wrongfully turned off in all the simulations, all the conclusions of the paper are potentially invalid.

We thank you for your attention to detail regarding our methodology. We have verified that bl_mynn_tkeadvect was indeed turned on and functional:

- Looking at the idealized, neutral 100TKE namelist for example (https://github.com/rybchuk/wfp-3dpblsensitivity/blob/main/runs/idealized/neutral/mynn-tke/namelist.input), we verify that "bl_mynn_tkeadvect = .true., .true.,"

- We have verified that the WRF code modifications from Issue 1235 are in the version of WRF that we use
- While TKE quickly recovers downwind of the turbines at the rotor disk heights, TKE is advected downwind above the rotor disk, most clearly seen in Figure 6a,b,d,e
- The prognostic version of the NCAR 3DPBL (which we use in all our simulations) inherently advects TKE (independently of bl_mynn_tkeadvect). The fact that the MYNN and the 3DPBL produce similar TKE addition plots futher suppors that MYNN has TKE advection turned on
- Finally, to definitively illustrate that TKE advection was turned on in the simulations in the manuscript, we have run a supplemental MYNN idealized neutral simulation in which the TKE advection was turned off. Figure RF1a has TKE advection turned on, and TKE is visibly advected downwind. Figure RF1b has TKE advection turned off, and TKE remains within vertical columns.

[Figure]

Response Figure 1. A TKE side view of idealized neutral simulations for which TKE advection is turned on (a) and turned off (b).

Thus, we are confident that our simulations in the manuscript were run with TKE advection turned on.

Minor points

1. 55: There is another wind farm parameterization for WRF in the literature: the hybrid model by Pan and Archer (2018).

We have updated the text to include a citation to Pan and Archer (2018) (L79).

2. 64 and 133 and 164: Missing citation "?"

Thank you. We have corrected the missing citations.

3. 101: I think I know what you are trying to say, but it needs to be defined better because an external wake cannot be defined as a "distance". Also, here you suse 0.2 m/s as the threshold, but in the rest of the paper it seems to be 0.5 m/s (e.g., Figure 3 and 11, dashed blue line).

Following feedback from Reviewer 1, the literature review section has been removed.

While the text has been removed, we want to address the challenge of characterizing external wakes here, as this feedback is also brought up later. As the recent WFP literature review paper by Fischereit et al. (2021) states

"One challenge that we identified is that from our review there is no standardized or common definition of a recovery length behind a farm. Studies used for instance the e-folding distance (Fitch et al. 2012), the location of 2% difference between a simulation with and without wind farms (Pryor et al. 2020) or the location where the wind speed has recovered to 95% of the freestream wind speed (Cañadillas et al. 2020). Due to this variety of different definitions, it is difficult to compare wake lengths across studies quantitatively."

Each of these studies selects a definition of an external wake that is reasonable for the question they are studying. In our analysis, we study wakes from large farms of large turbines. As such, we select three metrics: 1 m/s threshold, 0.5 m/s threshold, and the e-folding distance because we expect large wakes. This is imperfect, but we also acknowledge this challenge that the field faces.

4. Table 1: the same label here is used to indicate three different runs. Please use unique labels for each run, like "S-NWF" for stable, "U-NWF" for unstable etc.

While it is entirely reasonable to label the idealized runs "S-NWF" and "U-NWF", it would not be reasonable to name the monthlong real run in a similar manner. Thus, for the sake of consistency between the ideal runs and the real runs, we retain the original label format.

5. 203: not OK to cite a manuscript in preparation, please remove Rosencrans et al.

We have removed the reference to Rosencrans et al.

6. 206: type for "pseudo"

We have corrected the typo, thank you for catching it.

7. 209: How many turbines are there in total? 25 perhaps?

There are 100 turbines in the idealized simulations, as denoted by "The second case (100TKE) includes a 10-by-10 grid turbines based on the of 12-MW International Energy Agency" (L186).

8. 322: Why 0.5 m/s deficit if 0.2 m/s was stated earlier?

As stated in response to Minor Comment 3, there is no standard for external wake characterization and 0.5 m/s threshold makes sense for the scale of problem we are studying.

9. 322: I cannot understand what the e-folding distance is. Please include an equation. To be honest, I do not even understand why this variable is even introduced, it does not add much, it is overly sensitive to the stability and choice of the scheme, and it is no longer used in the real simulations later. Consider dropping it since it does not add much.

The e-folding distance is a relative measure used to characterize external wakes. The metric was introduced in the original Wind Farm Parameterization paper (Fitch et al., 2012), and as our idealized simulations parallel much of the analysis in that paper, we feel that it is appropriate to include the metric. We provide a formula on how to calculate the metric: "we calculate the e-folding contour as 1/e times the average internal wake strength, or about 36% (Fitch et al., 2012)" (L289)

10. Figure 3: I am surprised that the maximum deficit possible is 1 m/s (note that the maximum deficit is 4 m/s in Figure 11). This must be the most efficient ideal wind farm ever designed. Why is the flow from the west-southwest? I would recommend using white for the range -0.25 – 0.25 m/s.

- The new, idealized offshore simulations show maximum deficits in excess of 2 m/s. The mid-Atlantic simulations show stronger deficits because (a) there are substantially more turbines (1418 in LEASE, 177 in VW-ONLY, and 100 in the idealized simulations) and (b) Vineyard Wind is longer with respect to the dominant wind direction, thus leading to stronger internal waking.
- The flow in the idealized simulations is from the west-southwest because of the combination of friction and the Coriolis effect. The same effect was observed in Fitch et al. (2012). We have updated the paper to mention this effect: "Wakes are rotated from the U-geostrophic wind due to the combination of friction and the Coriolis force." (Figure 4 caption).
- We retain the original colorbar values for the range –0.25 - 0.25 m s-1 so that the subtle but real acceleration in the stable idealized simulations is visualized.

11. 345-350: I find it **very** difficult to believe that the addition of TKE causes a longer wake. Also very confusing that the weird decrease in TKE in one specific case (Figure 6g) can be used to explain this general and counter-intuitive finding. To me this is another flag that suggests that advection of TKE was **not** turned on.

Please see our response to Major Comment 2 for analysis that demonstrates that TKE advection was turned on.

12. 384: not OK to cite a manuscript under review. Please remove Bodini et al.

Bodini et al. has been published during the review of this manuscript, and the reference has been updated accordingly.

13. 409: the authors themselves note that there is no advection of TKE! This is not a realistic result. Flag bl_mynn_tkeadvect must be true for TKE to be advected, at least with the MYNN scheme.

We state that "tend[s] to not advect *far* downwind" (emphasis mine). Please see our response to Major Comment 2 for analysis that shows that TKE advection was turned on.

14. Figure 8: please use one color scheme! You can intervals that are variable to better emphasize features, but using two colorbars like that is not OK.

The updated idealized simulations all have similar hub-height wind speeds, and correspondingly, similar power production. Figure 8 now only uses one color scheme.

15. 476: Are these results with 0% TKE or 100% TKE? Why not 25% TKE as recommended?

These results are with 100% TKE. Thank you for noting that omission, and we have updated the Methods section to include that detail. "All wind plant simulations are run with $\alpha=1$. While validation of this parameter is limited, we note that Larsen and Fischereit (2021) saw more accurate results in an offshore wake study with that value ($\alpha=1$) than the value of $\alpha=0.25$ recommended by Archer et al. (2020)." (L232-234)

16. 484: define "centroid"

We have updated the text to read "we calculate average profiles at the middle ofVineyard Wind 1."

17. Figure 12: as in #14, not OK to have 4 colorbars.

Figures with multiple colorbars are employed within academic literature. For example, see Figure 3 in Pryor et al. (2020) and Figure 7 in Brugger et al. (2022).

18. 28: by this point, I could not force myself to read the manuscript anymore. Too boring and pointless. This section on the real cases is rather useless without observations and does not add anything to the discussion of the idealized cases. The paper would be better off without Section 4.

We hope the reviewer will be willing to complete its review now that we have clarified some points (e.g., that TKE advection was indeed turned on) above and that the manuscript has been improved. As we would like to stress again, our U.S. east coast analysis reveals important information that a number of stakeholders (scientists, government planners, industry) care about. Perhaps the most important and the most simple finding of the real analysis can be found in our abstract: "the 3DPBL generates 4.7%-7.8% less power than MYNN in August 2020, depending on the turbine build-out scenario". Our study shows that the choice of PBL scheme could lead to significantly different AEP predictions, and in theory, the choice of wind farm could flip AEP from being in a profitable scenario to an unprofitable scenario.

---

## Referee Report (RR1)

**Second review of "The Sensitivity of the Fitch Wind Farm Parameterization to a Three-Dimensional Planetary Boundary Layer Scheme" by Rybchuk et al., submitted to Wind Energy Science Discussions**

The authors have made concrete efforts to improve the paper and address the reviewers' concerns, including my own. I am especially glad that the authors checked the TKE advection issue and that they showed strong evidence that it was indeed active in their runs. That was my major reason of concern and it has been satisfactorily addressed.

As for the extremely long reply about the revisited motivation of the study to address uncertainty quantification (as opposed to validation), I could counter-argue that what the authors did is not exactly what Archer et al. (2014) recommended, which was basically to use ensembles. They meant several ensemble members, like 15-20, but here there are only 2 (MYNN and 3DPBL schemes). A 2-member ensemble is not sufficient to characterize uncertainty, but I will grant that the authors found a substantial difference in power, which is interesting and good to know, thus I will let this go.

**Replies to previous comments**

3. *101: I think I know what you are trying to say, but it needs to be defined better because an external wake cannot be defined as a "distance". Also, here you use 0.2 m/s as the threshold, but in the rest of the paper it seems to be 0.5 m/s (e.g., Figure 3 and 11, dashed blue line).*

> *Following feedback from Reviewer 1, the literature review section has been removed. While the text has been removed, we want to address the challenge of characterizing external wakes here, as this feedback is also brought up later. As the recent WFP literature review paper by Fischereit et al. (2021) states*
> *"One challenge that we identified is that from our review there is no standardized or common definition of a recovery length behind a farm. Studies used for instance the e-folding distance (Fitch et al. 2012), the location of 2% difference between a simulation with and without wind farms (Pryor et al. 2020) or the location where the wind speed has recovered to 95% of the freestream wind speed (Cañadillas et al. 2020). Due to this variety of different definitions, it is difficult to compare wake lengths across studies quantitatively."*
> *Each of these studies selects a definition of an external wake that is reasonable for the question they are studying. In our analysis, we study wakes from large farms of large turbines. As such, we select three metrics: 1 m/s threshold, 0.5 m/s threshold, and the e-folding distance because we expect large wakes. This is imperfect, but we also acknowledge this challenge that the field faces.*

> I think that the authors misunderstood my comment. I did not criticize the use of a threshold at all, that was and is just fine. I just found it confusing that the authors used two different values (0.2 m/s and 0.5 m/s) in two parts of the same manuscript. I

actually wonder if 0.2 was a typo perhaps? In fact, you no longer use 0.2 m/s in the revised manuscript. Not important anyway, but I thought I would clarify what I meant.

4. *Table 1: the same label here is used to indicate three different runs. Please use unique labels for each run, like "S-NWF" for stable, "U-NWF" for unstable etc.*

*While it is entirely reasonable to label the idealized runs "S-NWF" and "U-NWF", it would not be reasonable to name the monthlong real run in a similar manner. Thus, for the sake of consistency between the ideal runs and the real runs, we retain the original label format.*

I am sorry, but I disagree. You must not use the same label to indicate different runs, period. The point of a label is that it identifies uniquely the run you are talking about. However, the labels are not even used in the manuscript, so this discussion is moot. I would just suggest that the column "Label" be actually renamed "PBL scheme" and the prefix "NWF" be removed because this table is already for no-farm simulations.

8. *322: Why 0.5 m/s deficit if 0.2 m/s was stated earlier?*

*As stated in response to Minor Comment 3, there is no standard for external wake characterization and 0.5 m/s threshold makes sense for the scale of problem we are studying.*

Again, I was not criticizing the value, but noting the contradiction of using two different values in two different parts of the manuscript. All resolved now since there is no longer a 0.2 m/s threshold.

15. *476: Are these results with 0% TKE or 100% TKE? Why not 25% TKE as recommended?*

*These results are with 100% TKE. Thank you for noting that omission, and we have updated the Methods section to include that detail. "All wind plant simulations are run with α=1. While validation of this parameter is limited, we note that Larsen and Fischereit (2021) saw more accurate results in an offshore wake study with that value (α=1) than the value of α=0.25 recommended by Archer et al. (2020)." (L232-234)*

Actually, I read Larsen and Fischereit (2021) very carefully and they did not reach any such conclusion in their paper. They even explicitly stated that "It remains inconclusive … what are the correct $C_{TKE}$ values to use; more measurements are needed for further investigation." The authors might be referring to Figure 14 in Larsen and Fischereit (2021), which shows very large TKE injection over the wind farm during one 2-hour flight, so large that none of the parameterizations, no matter which settings were used, could capture it. In particular, the fact that not even the results with advection turned off, which are well known to cause an overshoot of TKE in the grid cells of the wind farm because TKE has no way to go but continue accruing, could match the observed values indicate that perhaps something was off with the methods used to calculate TKE from the measurements. Even

Larsen and Fischereit (2021) themselves did not seem to trust the findings enough to make any recommendations about C_TKE based on their study. Anyway, I do not intend to conduct a review of Larsen and Fischereit (2021) here, but I think I provided enough evidence to suggest that the sentence at lines 232-234 should be removed.

*17. Figure 12: as in #14, not OK to have 4 colorbars.*

*Figures with multiple colorbars are employed within academic literature. For example, see Figure 3 in Pryor et al. (2020) and Figure 7 in Brugger et al. (2022).*

While other papers may have had good reasons to make such a choice, I really do not see the advantage of 4 colorbars here. It is extremely hard to compare results from one heatmap to the next. The only advantage is that one could discern patterns within the wind farm, but the authors do not actually discuss any such patterns in the text, so there is really no advantage.

**New comments**

1. P. 3, l. 79: the paraterization by Abkar and Porte-Agel does not have a name and calling it the "Abkar WFP" is presumptuous. The parameterization by Pan and Archer (2018) also is not called the Pan WFP, but it has a name, it is called the "hybrid" WFP.
2. P. 3, l. 89: your research question has changed to "How sensitive are modeled mesoscale wakes to the choice of PBL parameterization?", which is a much better fit to the study, but I think you should be a bit more humble and admit that you have only examined one additional PBL scheme, one that has not really been validated much because it has not even been released officially with WRF as far as I know. So I would recommend that you add a few sentences to explain your choices along the lines of: "Ideally, we should test all 13 PBL schemes with the Fitch WFP to fully characterize the uncertainty. Here we propose, as a first step, to compare two PBL schemes: the conventional NYMM and the newly-developed NCAR 3DPBL. We chose the latter because …. " and explain why you chose it over the other 13 schemes? A few sentences suffice, like the fact that it has a prognostic equation for TKE.
3. P. 6, l. 168: good idea to match the hub height wind speed rather than the geostrophic speed.
4. P. 9, l. 233-235: as explained above, this sentence should be removed because it is not consistent with the recommendations and findings reported by Larsen and Fischereit (2021) themselves.
5. What is "pp"? It is used only in the last sections of the paper (that is why I did not notice I earlier) but is not defined. Perhaps it means just "%", which was used in the abstract and other sections? Please use one convention only.

---

## Author Response (AR2)

Dear authors,

Thanks a lot for your revised version. I can see you have undertaken an extensive revision of the manuscript and shorten it. I understand the main author is a PhD student and he is obviously dealing with a complex subject. But the manuscript lacks focus and having both idealized and real simulations together does not help. As the other reviewer and I mentioned in the previous review, the real simulations do not help the manuscript, the understanding of the physics/parametrizations, and makes it long. The real simulations need to be erased from the manuscript in order to move further. They do not need to be wasted; the PhD student can use them in another manuscript (if appropriate) or in his thesis dissertation. In the following, you find a list of major and minor/specific comments.

Dear reviewer, thank you for your continued thoughtful feedback on our manuscript. We have updated our manuscript accordingly. We will also note that as the review process has been going on, we have found and fixed the bug regarding strange near-surface TKE values in the idealized 3DPBL simulations. Thankfully, as we anticipated and noted earlier in the review, this bug had negligible effects on wind speed and TKE profiles.

Major comments:

1. Real simulations: There is no need for them. They do not clarify the role of the PBL scheme on the results. They do not clarify the role of the parametrization on the results either. Having them makes the manuscript long, and unfocused.

Following feedback from Reviewer 1 as well as the editor, we have entirely struck the presentation of the real simulations from the manuscript. However, following suggestions from Reviewer 1, we cite the lead author's dissertation and briefly discuss the results of the real simulations in the Conclusions (L360).

2. If the real-simulations are dropped then you can perform the idealized simulations with the full 3DPBL model, instead of using the "PBL-approximation". This will be definitively interesting when comparing the PBL schemes.

While we agree that analysis with the full 3DPBL model would be interesting, that analysis is beyond the scope of this manuscript.

3. Why are the idealized simulations setup with such high domains (20 km)? This is probably the reason why your simulations take so long to spin up. Why not using 2 to 5 km?

In our idealized simulations, we do our best to replicate the original Fitch et al. (2012) WFP study. That study had its model top set at 20 km (page 3021 of the paper), and as such, we use the same height.

4. Figure 3: Why is for the stable simulation TKE not linearly decreasing with height for the 3DPBL compared to MYNN? Why are the rotor height TKE values so different between the two PBL schemes under unstable and stable conditions? Why is the unstable TKE so different and low in MYNN compared to 3DPBL (it is even half the value of the stable simulation!)? Are the differences in the temperature profile close to the surface related to the way the PBL scheme is using the surface flux condition? Close to the surface MYNN seems more unstable (based on the temp. profile) than 3DPBL, yet the TKE is much lower for MYNN. Why are the results averaged over a 24-hour period (why not using 1 or 2 hours)?

In general, WRF simulations with different PBL schemes can produce different states of the atmosphere for a wide variety of reasons. However, in the case of pseduo-steady idealized simulations, it becomes much simpler to attribute the source of differences in atmospheric state. Here, the differences between MYNN and the 3DPBL primarily arise from different choices of closure constants and length scale formulations (L122). Recent studies using 3DPBL (Arthur et al. 2022; Juliano et al. 2022) suggest that model results under both stable and convective conditions are quite sensitive to these choices. Model sensitivity to these factors is actively being explored (Eghdami et al. (2022)). Thus, addressing each of these questions point-by-point:
- The difference in stable TKE profiles between MYNN and the 3DPBL likely arises because the 3DPBL and MYNN have different closure constants and length scale formulations.
- Hub-height TKE values are different between the two PBL schemes in stable and unstable conditions because we designed our experimental set up to match hub-height wind speeds instead of TKE values.
- As was the case in the stable simulations, the TKE profiles are different in the unstable simulations likely because of the different closure constants and length scale formulations.
- Temperature profiles are the same for MYNN and the 3DPBL in neutral and unstable conditions. They differ for stable conditions because the two PBL schemes were spun up for different durations, in order for them to reach matching hub-height wind speeds (Figure 1).
- As you have noted below in the minor comments, MYNN exhibits unexpected behavior in its unstable TKE profile---it unexpectedly produces more TKE in neutral conditions than in unstable conditions. The fact that stronger near-surface temperature gradients in MYNN lead to less TKE than in the 3DPBL is likely driven by this same confounding mechanism. On top of this, again the difference in closure constants and length scales will further drive the PBL schemes to predict different near-surface TKE levels.
- We chose to average over a 24 hour period because oscillations of hub-height wind speed lasted about 12 hours (Figure 1), and we wanted to smooth those out.

Minor comments:

1. Line 38 delete "more"
We have removed "more", and the line now went from "While this approach is more common for onshore sites" to "While this approach is more common for onshore sites".

2. Line 54 it should be "planetary"

Thank you for catching our typo. We have corrected it.

3. Line 57 are the traditional PBL schemes governing vertical turbulence fluxes or all turbulence fluxes (as it reads)

We have clarified the statement from "PBL schemes govern turbulent fluxes and mixing within the atmospheric boundary layer" to "PBL schemes govern turbulent fluxes (typically just vertical turbulent fluxes) and mixing within the atmospheric boundary layer" (L52-53).

4. Line 70 it should read "in order to"

Thank you for catching our typo. We have corrected it.

5. Lines 78-80 the sentence reads as NWP models use these WFPs, but is there a NWP model that uses Abkar's?

Thank you for bringing this to our awareness. The WFP review paper (Fischerit et al. 2021) refers to the "Abkar WFP". However, we have not found an NWP paper that uses this WFP, so we have removed the reference to it.

6. Line 83 Replace "that some" by "that most if not all"

We have replaced "that some" with "that most if not all".

7. Lines 127-128 Why do the constants of the 3DPBL come from MY82 and not from the updated NN89?

The choice of closure constants is an active area of research for the 3DPBL. In this work, we selected the MY82 closure constants as they were tested in Juliano et al. (2022) whereas the NN89 constants were not tested.

8. Line 148. You need to explain why not using Archer's suggestion for \alpha

We initially had an explanation for our choice in \alpha in the Methodology section for the real simulations, but the other reviewer told us to cut the explanation, ultimately because "Larsen and Fischereit (2021) themselves did not seem to trust the findings enough to make any recommendations about C_TKE based on their study". Thus, following the other reviewer's suggestion, we are not explaining our choice.

9. Table 1. There is a problem with the units of the heat flux

Thank you for catching our typo. We have corrected it.

10. Lines 170-176 All this wording can be omitted. I guess the real reason why you wanna match the hub height wind speed is to have the same Ct values

We have removed Lines 170-176 to make the paragraph more concise, thank you.

11. Line 180 Does it make sense to force sable simulations imposing surface fluxes? See Basu et al. (2008)

This is a very fair question. We chose to force the idealized stable simulations with a heat flux so that they could be compared to "typical" conditions in the offshore real simulations. This was

helpful for the comparison, as it is much stranger to try and calculate a "representative cooling rate" of the real simulations. In a context where real simulations are not considered, we agree that a temperature cooling rate would be preferred to a heat flux at the surface. However, reading Basu et al. (2008), we believe our simulations here are okay, because our simulations are only weakly stable. The conclusion of Basu et al. (2008) states "It is argued that any PBL model (single column or LES) will only be able to capture the near-neutral to weakly stable regime if surface sensible heat flux is prescribed... In order to represent the moderate to very stable regime in a boundary layer model, unquestionably one needs to use surface temperature as a boundary condition as shown in this paper." Because we are weakly stable, using heat flux should be acceptable.

12. Line 182 How do you estimate the height of the boundary layer?
We now clarify, "After spin up, the boundary-layer height as determined through the NWF temperature profile (Fig. 2) is approximately 250 m in the stable simulations, 550 m in the neutral simulations, and 600 m in the unstable simulations." (L172-173)

13. Line 242 well there is no real "turbulent hub height wind speed" in a PBL simulation
We have removed the word "turbulent" in order to be more precise.

14. Line 245 These spin ups will be much shorter if using lower top boundaries; also you will have perhaps less numerical noise and unwanted waves
While this may be true, we retain the tall upper boundary in order to be consistent with Fitch et al. (2012).

15. Lines 246-249 These lines can be omitted
Thank you for noting this. We elected to retain these lines, as we think the transition sentences are helpful for the reader so there is clear motivation on why we need to discuss NWF profiles before discussing wakes.

16. From the results in Fig. 2 it does not seem like the stable and unstable simulations stabilize
Indeed, the hub-height wind speed continues to change for the stable and unstable simulations, whereas it is near-constant for the neutral simulations. However, this non-stationarity is basically an unavoidable part of studying the ABL in stable and unstable conditions. For example, as long as we continue to apply a positive heat flux, the height of the capping inversion will continue to grow in the unstable ABL. Our averaged hub-height wind speeds are "pseudo-stationary" however, and as such, we believe they are sufficient to study averaged wake effects.

17. Line 266 Any idea why the TKE is weaker in the unstable than in the neutral MYNN? Could something be wrong?
This phenomena is something we have struggled with. MYNN is a thoroughly tested and well-established PBL scheme, so we were surprised to see this behavior as well. We are confident that our simulation inputs are correct (and they can be double checked through our open-sourced code).

18. Line 286 something is weid in "should be run that are"

Thank you for catching our strange wording. We have reworded "Thus, this variability within the idealized runs suggests that real-world case studies should be run that are tailored to a specific region and turbine configuration." to "Thus, this variability within the idealized runs suggests that real-world case studies should be tailored to a specific region and turbine configuration.".

19. Line 288 "external wakes" was introduced when you introduce the real cases but now you use it for idealized cases… which do not have external wakes… also line 296

Thank you for noting this inconsistency. We have updated the text to now use terminology like "the wake within the extent of the plant" and "the wake outside the extent of the plant".

20. Figure 5 Is the side view a vertical cross section parallel to x? and where in y?

This specifically a side view of horizontally averaged wind speed deficits parallel to x. Deficits are averaged over the y-extent of the farm (L265).

21. Caption Fig. 6: it should read "…6a was calculated as the difference between the results in panels…" Also you say "are calculated in a similar manner": this is not really true because you do not have tendencies in Fig. 5

We have updated the text from reading "Figure 5a was calculated as panel Figs. 4a-4g." to "Figure 5a was calculated as the difference between the results in panels Figs. 4a-4g." (caption for Figure 5). We have also updated the following sentence from "The u-tendency deficits in the 100TKE simulations are calculated in a similar manner." to "The u-tendency deficits in the 100TKE simulations are calculated using a similar procedure involving tendencies."

22. Line 163: this is interesting. Why Mangara et al. saw substantial changes in TKE so far downwind? There is no mechanism to preserve TKE so far I think

We are also surprised by the Mangra et al. results. Both our idealized and real simulations showed that TKE was advected a much shorter distance than what they saw. The real simulations of Siedersleben et al. (2020) also show TKE advection over a much shorter distance than Mangara et al. found.

23. Lines 364-366 But this observation is because in stable and unstable conditions the TKE in the NWF is nearly doubled in 3DPBL than MYNN?

Not exactly. While yes the 3DPBL produces more TKE than MYNN in stable and unstable conditions for the NWF scenario, the two PBL schemes produce near-identical TKE profiles in the neutral case. That is why L364-366 read "The 100TKE 3DPBL simulations also *consistently* predict stronger levels of additional TKE than their MYNN counterparts" (emphasis here only).

24. Figure 7: see my comment 20

To be clear in the consistency between figures, we have updated this caption to read "Same as Fig. 5, but for TKE..."

25. Figure 8: Why are the capacity factors so different for the turbines on the left of the array for neutral conditions compared to stable and unstable? They should be nearly the same at those turbines right?

This is a good point, thank you for catching it. These differences in the arise ultimately show up in the visualization because power production is extremely sensitive to velocity, following $U^3$. This is why power production even strictly within the leftmost column of any one simulation can substantially vary. For example, Fig 7a has capacity factor ranging between 54-66% in the upwind column. This sensitivity is then coupled with local variations in wind speed. The neutral wind speeds were fairly constant during the 24 hour performance period (Fig 1), whereas the stable and unstable winds were more variable. Because power production scales with $U^3$, the stronger variability in time results in an over-exaggeration of power production, even though all simulations have a 24-hour average wind speed near 9.35 m/s.

26. Lines 411-412 "The unstable… are 7.69 m/s… NWF profiles". This is not needed.

We have entirely cut this part of the text, as the real simulations have been removed.

27. Figure 9: in neutral conditions, you most probably have the lowest friction velocities, which translates also in the lowest winds compared to the other stability cases. And this is kind of a problem as your stable case is probably biased by a large number of near-neutral cases; cases that would have fall into the category of neutral if the Obukhov length (instead of the flux only) was used.

As was the case for the previous comment, this section has been removed because the real simulations are gone.

28. Line 535: "MYNN predicted strong….. in stable conditions" This was not the case for the mid atlantic simulations, why? Maybe related to comment 27?

We note that stable MYNN simulations produced longer wakes than the 3DPBL in the ideal case (Fig 3 a,g) as well as in the real case (old Fig 11 a,g). However, the real simulations have been removed, so this point is moot.

References
Basu et al. (2008) An inconvenient "truth" about using sensible heat flux as a surface boundary condition in models under stable stratified regimes. Acta Geophysica, 56, 88—99

References
- Arthur, R. S., et al. "Improved Representation of Horizontal Variability and Turbulence in Mesoscale Simulations of an Extended Cold-Air Pool Event, Journal of Applied Meteorology and Climatology" 2022, 61(6), 685-707.
- Eghdami, Masih, et al. "Diagnosis of Second-Order Turbulent Properties of the Surface Layer for Three-Dimensional Flow Based on the Mellor–Yamada Model." Monthly Weather Review, vol. 150, no. 5, May 2022, pp. 1003–21. journals.ametsoc.org, https://doi.org/10.1175/MWR-D-21-0101.1.

- Fitch, Anna C., et al. "Local and Mesoscale Impacts of Wind Farms as Parameterized in a Mesoscale NWP Model." Monthly Weather Review, vol. 140, no. 9, Sept. 2012, pp. 3017–38. journals.ametsoc.org, https://doi.org/10.1175/MWR-D-11-00352.1
- Fischereit, Jana, et al. "Review of Mesoscale Wind-Farm Parametrizations and Their Applications." Boundary-Layer Meteorology, Aug. 2021. Springer Link, https://doi.org/10.1007/s10546-021-00652-y.
- Juliano, Timothy W., et al. "'Gray Zone' Simulations Using a Three-Dimensional Planetary Boundary Layer Parameterization in the Weather Research and Forecasting Model." Monthly Weather Review, vol. 1, no. aop, Oct. 2022. journals.ametsoc.org, https://doi.org/10.1175/MWR-D-21-0164.1.
- Siedersleben, Simon K., et al. "Turbulent Kinetic Energy over Large Offshore Wind Farms Observed and Simulated by the Mesoscale Model WRF (3.8.1)." Geoscientific Model Development, vol. 13, no. 1, Jan. 2020, pp. 249–68. gmd.copernicus.org, https://doi.org/10.5194/gmd-13-249-2020.

**Second review of "The Sensitivity of the Fitch Wind Farm Parameterization to a Three Dimensional Planetary Boundary Layer Scheme" by Rybchuk et al., submitted to Wind Energy Science Discussions**

The authors have made concrete efforts to improve the paper and address the reviewers' concerns, including my own. I am especially glad that the authors checked the TKE advection issue and that they showed strong evidence that it was indeed active in their runs. That was my major reason of concern and it has been satisfactorily addressed. As for the extremely long reply about the revisited motivation of the study to address uncertainty quantification (as opposed to validation), I could counter-argue that what the authors did is not exactly what Archer et al. (2014) recommended, which was basically to use ensembles. They meant several ensemble members, like 15-20, but here there are only 2 (MYNN and 3DPBL schemes). A 2-member ensemble is not sufficient to characterize uncertainty, but I will grant that the authors found a substantial difference in power, which is interesting and good to know, thus I will let this go.

Dear reviewer, thank you for your continued thoughtful feedback on our manuscript. We have updated the manuscript in accordance with your suggestions. We note that following feedback from the editor and Reviewer 1 in this latest round of revisions, we have cut nearly everything regarding the real simulations. We will also note that as the review process has been going on, we have found and fixed the bug regarding strange near-surface TKE values in the idealized 3DPBL simulations. Thankfully, as we anticipated and noted earlier in the review, this bug had negligible effects on wind speed and TKE profiles.

**Replies to previous comments**
*3. 101: I think I know what you are trying to say, but it needs to be defined better because an external wake cannot be defined as a "distance". Also, here you use 0.2 m/s as the threshold, but in the rest of the paper it seems to be 0.5 m/s (e.g., Figure 3 and 11, dashed blue line).*

*Following feedback from Reviewer 1, the literature review section has been removed. While the text has been removed, we want to address the challenge of characterizing external wakes here, as this feedback is also brought up later. As the recent WFP literature review paper by Fischereit et al. (2021) states*
*"One challenge that we identified is that from our review there is no standardized or common definition of a recovery length behind a farm. Studies used for instance the efolding distance (Fitch et al. 2012), the location of 2% difference between a simulation with and without wind farms (Pryor et al. 2020) or the location where the wind speed has recovered to 95% of the freestream wind speed (Cañadillas et al. 2020). Due to this variety of different definitions, it is difficult to compare wake lengths across studies quantitatively."*
*Each of these studies selects a definition of an external wake that is reasonable for the question they are studying. In our analysis, we study wakes from large farms of large turbines. As such, we select three metrics: 1 m/s threshold, 0.5 m/s threshold, and the efolding distance because we expect large wakes. This is imperfect, but we also acknowledge this challenge that the field faces.*

I think that the authors misunderstood my comment. I did not criticize the use of a threshold at all, that was and is just fine. I just found it confusing that the authors used two different values (0.2 m/s and 0.5 m/s) in two parts of the same manuscript. I actually wonder if 0.2 was a typo perhaps? In fact, you no longer use 0.2 m/s in the revised manuscript. Not important anyway, but I thought I would clarify what I meant.

*Thank you for clarifying.*

*4. Table 1: the same label here is used to indicate three different runs. Please use unique labels for each run, like "S-NWF" for stable, "U-NWF" for unstable etc.*

*While it is entirely reasonable to label the idealized runs "S-NWF" and "U-NWF", it would not be reasonable to name the monthlong real run in a similar manner. Thus, for the sake of consistency between the ideal runs and the real runs, we retain the original label format.*

I am sorry, but I disagree. You must not use the same label to indicate different runs, period. The point of a label is that it identifies uniquely the run you are talking about. However, the labels are not even used in the manuscript, so this discussion is moot. I would just suggest that the column "Label" be actually renamed "PBL scheme" and the prefix "NWF" be removed because this table is already for no-farm simulations.

*Thank you for the feedback. We have updated Table 1 so that the column "Label" is now called "PBL Scheme" and we have removed the "NWF" prefix.*

*8. 322: Why 0.5 m/s deficit if 0.2 m/s was stated earlier?*

*As stated in response to Minor Comment 3, there is no standard for external wake characterization and 0.5 m/s threshold makes sense for the scale of problem we are studying.*

Again, I was not criticizing the value, but noting the contradiction of using two different values in two different parts of the manuscript. All resolved now since there is no longer a 0.2 m/s threshold.

15. 476: Are these results with 0% TKE or 100% TKE? Why not 25% TKE as recommended?

*These results are with 100% TKE. Thank you for noting that omission, and we have updated the Methods section to include that detail. "All wind plant simulations are run with α=1. While validation of this parameter is limited, we note that Larsen and Fischereit (2021) saw more accurate results in an offshore wake study with that value (α=1) than the value of α=0.25 recommended by Archer et al. (2020)." (L232-234)*

Actually, I read Larsen and Fischereit (2021) very carefully and they did not reach any such conclusion in their paper. They even explicitly stated that "It remains inconclusive ... what are the correct C_TKE values to use; more measurements are needed for further investigation." The authors might be referring to Figure 14 in Larsen and Fischereit (2021), which shows very large TKE injection over the wind farm during one 2-hour flight, so large that none of the parameterizations, no matter which settings were used, could capture it. In particular, the fact that not even the results with advection turned off, which are well known to cause an overshoot of TKE in the grid cells of the wind farm because TKE has no way to go but continue accruing, could match the observed values indicate that perhaps something was off with the methods used to calculate TKE from the measurements. Even Larsen and Fischereit (2021) themselves did not seem to trust the findings enough to make any recommendations about C_TKE based on their study. Anyway, I do not intend to conduct a review of Larsen and Fischereit (2021) here, but I think I provided enough evidence to suggest that the sentence at lines 232-234 should be removed.

*Following feedback from everyone involved in the peer-review process, we have cut any mention of the real simulations, and as such, L232-234 have been removed.*

17. Figure 12: as in #14, not OK to have 4 colorbars.
*Figures with multiple colorbars are employed within academic literature. For example, see Figure 3 in Pryor et al. (2020) and Figure 7 in Brugger et al. (2022).*

While other papers may have had good reasons to make such a choice, I really do not see the advantage of 4 colorbars here. It is extremely hard to compare results from one heatmap to the next. The only advantage is that one could discern patterns within the wind farm, but the authors do not actually discuss any such patterns in the text, so there is really no advantage.

As was the case for the above comment, nearly all mentions of real simulations have been cut from the manuscript, and as such, Figure 12 and its 4 colorbars have been removed.

**New comments**

1. P. 3, l. 79: the paraterization by Abkar and Porte-Agel does not have a name and calling it the "Abkar WFP" is presumptuous. The parameterization by Pan and Archer (2018) also is not called the Pan WFP, but it has a name, it is called the "hybrid" WFP.

We called it the Abkar WFP following Fischeit et al. (2021), but we have removed mention of this WFP following feedback form the other reviewer. We have now also updated "Pan WFP" to the "hybrid WFP".

2. P. 3, l. 89: your research question has changed to "How sensitive are modeled mesoscale wakes to the choice of PBL parameterization?", which is a much better fit to the study, but I think you should be a bit more humble and admit that you have only examined one additional PBL scheme, one that has not really been validated much because it has not even been released officially with WRF as far as I know. So I would recommend that you add a few sentences to explain your choices along the lines of: "Ideally, we should test all 13 PBL schemes with the Fitch WFP to fully characterize the uncertainty. Here we propose, as a first step, to compare two PBL schemes: the conventional NYMM and the newly-developed NCAR 3DPBL. We chose the latter because …. " and explain why you chose it over the other 13 schemes? A few sentences suffice, like the fact that it has a prognostic equation for TKE.

We wholeheartedly agree with your framing and we have updated the closing paragraph of the introduction accordingly (edits in italics): "In this paper, *we begin to* address the question: How sensitive are modeled mesoscale wakes to the choice of PBL parameterization? *Ideally, this question would be addressed by studying all 13 PBL schemes in WRF with the Fitch WFP insofar as that is possible. Here, as a first step, we compare two PBL schemes*: MYNN (Nakanishi and Niino 2009) and the recently developed NCAR 3DPBL (Kosovic et al. 2020, Juliano et al 2022). *We chose the latter as it as a prognostic equation for TKE, which is important as the Fitch WFP modifies TKE fields.*"

3. P. 6, l. 168: good idea to match the hub height wind speed rather than the geostrophic speed.

Thank you.

4. P. 9, l. 233-235: as explained above, this sentence should be removed because it is not consistent with the recommendations and findings reported by Larsen and Fischereit (2021) themselves.

This sentence has been removed.

5. What is "pp"? It is used only in the last sections of the paper (that is why I did not notice I earlier) but is not defined. Perhaps it means just "%", which was used in the abstract and other sections? Please use one convention only.

"pp" is percentage points (defined in L224). It is used in early parts of the paper to take the difference in wake wind speed deficit percentages (e.g. 14% - 12% is 2pp) as well as later to discuss differences in capacity factor, which is also expressed as a percentage.

---

## Author Response (AR3)

Dear authors,

I think the paper now reads very well and I would like to thank you for considering many of the comments I proposed. I only have two minor comments that I need you to consider because I think it is only fair to do so:

Dear reviewer,

Thank you for your continued help on this manuscript. We have integrated your feedback into the manuscript and summarize our changes below.

1 Line 79: you are being politely by saying "the results from these studies should be interpreted with caution". I am sorry but in this case one cannot be polite. The bug in the code has enormous implications and surely all results from previous studies are wrong. So it is not about interpretation; it is about providing results and conclusions that are wrong.

Thank you for underscoring the severity of this bug. As far as we are aware, the effects of this bug have only been studied in one steady-state, idealized neutral simulation for one background wind speed. Without additional analysis, we are hesitant to make claims about impacts in idealized simulations in unstable or stable simulations, as well as real simulations. However, we agree that the magnitude of the impact of this bug may be large, and we have updated the manuscript to reflect this sentiment. The text now reads "the results from these studies should be interpreted with caution, as it is possible that this bug may have significant impacts." (L80).

2. Line 213: "TKE are weaker in the unstable MYNN..." They are not weaker; they are more than halved! They are also half the values of the stable ones. Here the reader would like to know if this is what one might expect or if this is an issue of the simulation. You do not really try to explain it and I am kind of ok with that but then I would like you to state something more... could it be the way you run the simulations (domain-wise)? could it be a bug in MYNN? could this be true and then 3DPBL be wrong? I think this point is quite important because Fitch explicitly introduces TKE at the turbine grid cell and so I would think that the inflow TKE is quite important for the understanding of the abilities of the scheme. You are obviously not able to match TKEs between the schemes (which is ok as they are different) but the issue here is that you have two completely different trends of TKE with stability with major differences in the values.

While we also find the behavior of MYNN's TKE odd, we are confident that we do not have a bug in our simulations, and we provided the namelist so that others may also investigate this behavior. Our current explanation for the weak TKE is that the MYNN constants were calibrated against the land-based Wangara experiment, which has substantially stronger heat flux values (as high as 0.21 K m/s, Nakanishi and Niino 2009) than the offshore-based heat flux that we use (20 W/m2 ~= 0.015 K m/s). We conclude the paragraph "Contrary to what might be expected, we note that hub-height values of TKE are weaker in the unstable MYNN simulations than in the

neutral MYNN simulations. We hypothesize that these low TKE values occur due to the weak heat fluxes." (L216)